# An Option-Dependent Analysis of Regret Minimization Algorithms in Finite-Horizon Semi-MDP

**Gianluca Drappo**
*Department of Electronics, Information and Bioengineering*
*Politecnico di Milano*                                    *gianluca.drappo@polimi.it*

**Alberto Maria Metelli**
*Department of Electronics, Information and Bioengineering*
*Politecnico di Milano*                                    *albertomaria.metelli@polimi.it*

**Marcello Restelli**
*Department of Electronics, Information and Bioengineering*
*Politecnico di Milano*                                    *marcello.restelli@polimi.it*

**Reviewed on OpenReview:** *https://openreview.net/forum?id=VP9p4u9jAo*

## Abstract

A large variety of real-world Reinforcement Learning (RL) tasks are characterized by a complex and heterogeneous structure that makes end-to-end (or flat) approaches hardly applicable or even infeasible. Hierarchical Reinforcement Learning (HRL) provides general solutions to address these problems thanks to a convenient multi-level decomposition of the tasks, making their solution accessible. Although often used in practice, few works provide theoretical guarantees to justify this outcome effectively. Thus, it is not yet clear when to prefer such approaches compared to standard flat ones. In this work, we provide an option-dependent upper bound to the regret suffered by regret minimization algorithms in finite-horizon problems. We illustrate that the performance improvement derives from the planning horizon reduction induced by the temporal abstraction enforced by the hierarchical structure. Then, focusing on a sub-setting of HRL approaches, the options framework, we highlight how the average duration of the available options affects the planning horizon and, consequently, the regret itself. Finally, we relax the assumption of having pre-trained options to show how, in particular situations, is still preferable a hierarchical approach over a standard one.

## 1 Introduction

Hierarchical Reinforcement Learning (HRL, Pateria et al., 2021) is a learning paradigm that decomposes a long-horizon Reinforcement Learning (RL, Sutton & Barto, 2018) task into a sequence of potentially shorter and simpler sub-tasks. The sub-tasks could be further divided, generating a hierarchical structure organized at arbitrary levels. Each of these defines a different problem, where the original action space is replaced by the set of sub-tasks available on the lower level, and the same could be replicated for multiple levels. Although, the actual state transition is induced only once the control reaches the leaf nodes, where the policies choose among the primitive actions (i.e., actions of the original MDP on top of which the hierarchy is constructed). For the higher levels, once a sub-task is selected, the control passes to the relative internal policy until its termination. This introduces the concept of *temporal abstraction* (Precup & Sutton, 1997), for what concerns the high-level policy, the action persists for a certain time, resulting in an actual reduction of the original planning horizon.

Several algorithms demonstrate outstanding performance compared to standard RL approaches in several long-horizon problems (Levy et al., 2019; Vezhnevets et al., 2017; Bacon et al., 2017; Nachum et al., 2018).

However, such evidence is mainly emerging in practical applications, and the theoretical understanding of the inherent reasons for these advantages is still underdeveloped. Only a few papers tried to justify these advantages theoretically, focusing on different aspects. For instance, Mann et al. (2015) studies the convergence of an algorithm that uses temporally extended actions instead of primitive ones. Fruit et al. (2017) and the extension Fruit & Lazaric (2017) focus on the exploration benefit of using options in average reward problems. Finally, more recently, Wen et al. (2020) show how the MDP structure affects regret.

In this paper, we seek to further bridge this theory-practice gap by analyzing the impact of temporal abstraction on the complexity of finite-horizon problems, a class in which the influence of a hierarchical structure naturally emerges. The complexity of solving a finite-horizon MDP depends on the length of the planning horizon. Intuitively, the longer the planning horizon, the more complex the problem. Introducing a hierarchy of sub-tasks temporally partitioned the problem into many different *potentially* shorter sub-problems. As mentioned above, once a level policy selects an action, the control passes to the lower level until the sub-task terminates. Typically, if the hierarchy is well constructed, a sub-task would execute for more than one primitive step, during which the high-level has neither control over what is happening, and often, nor visibility. Thus, concerning a policy that selects among sub-tasks, the number of actions (sub-tasks in this case) played in one episode depends on the duration of the available sub-tasks and is smaller than the original amount. This procedure produces an actual planning horizon reduction, scaled down by this term proportional to the sub-tasks duration, resulting in a less complex problem than the original one.

Firstly, we extend the formalism of FH-MDP to include temporally extended actions, defining an FH-SMDP. Then, we show that standard algorithms for FH-MDPs, can not be used to find a solution in this new setting, being unable to treat temporally extended actions properly. Therefore, we propose an approach, FH-SMDP-UCRL, to solve Finite-Horizon Semi-Markov Decision Processes when a set of fixed pre-trained sub-tasks policies are given (*options*) to theoretically demonstrate this effective improvement in complexity compared to a standard approach for Finite-Horizon MDPs. Moreover, to move toward a more realistic application of Hierarchical methods, we provide insights to characterize problems in which the benefits above persist even when no fixed policies are available. We present an Explore-Then-Commit approach (Lattimore & Szepesvári, 2020), that first separately learns each sub-task policy and then exploits FH-SMDP-UCRL to solve the SMDP with fixed policies. Afterward, we theoretically prove that this second approach would also benefit performance in specific MDPs compared to a standard approach. Eventually, both these results could help to discriminate among problems in which a hierarchical approach could be more effective than a standard one.

**Contributions** The paper's contributions can be summarized as follows. (1) We propose a new algorithm for the finite-horizon setting that exploits a set of *fixed* options (Sutton et al., 1999) to solve the HRL problem (Section 4). (2) We conducted a regret analysis of this algorithm, providing an option-dependent upper bound, which, to the best of our knowledge, is the first in the Finite Horizon setting. This result could be used to define new objective functions for options discovery methods to search for options minimizing this regret (Section 4.1). (3) For the sake of our analysis, we formulate the notion of Finite Horizon SMDP and a *performance difference lemma* for this setting (Section 3, 7). (4) We conducted a finer analysis to bound the expected number of temporally extended actions selected in one episode, resorting to the renewal process theory (Section 5). (5) Lastly, we extend our analysis to the options policies learning; we present an Explore-Then-Commit approach and characterize a class of problems more efficiently solvable through a Hierarchical approach, even when no set of pre-trained policies is available (Section 6). Therefore, we demonstrate that there are problems for which better guarantees in terms of sample complexity are achieved when solved by a hierarchical approach when both a set of fixed options are provided and not.

## 2 Preliminaries and Problem Definition

In this section, we provide the necessary background that will be employed in the remainder of the paper.

**Finite-Horizon MDPs** A Finite Horizon Markov Decision Process (MDP, Puterman, 2014) is defined as a tuple $\mathcal{M} = (\mathcal{S}, \mathcal{A}, p, r, H)$, where $\mathcal{S}$ and $\mathcal{A}$ are the finite state and the primitive action spaces, respectively, $p(s'|s, a, h)$ is the transition probability function defining the probability of transitioning to state $s' \in \mathcal{S}$ by

taking action $a \in \mathcal{A}$ in state $s \in \mathcal{S}$ at stage $h \in [H]$. $r(s, a, h)$ is the reward function that evaluates the quality of action $a \in \mathcal{A}$ when taken in state $s \in \mathcal{S}$ at stage $h \in [H]$, and $H$ is the horizon, which defines the duration of each episode of interaction with the environment. The behavior of an agent is modeled by a deterministic policy $\pi : \mathcal{S} \times [H] \to \mathcal{A}$ that maps states $s \in \mathcal{S}$ and stages $h \in [H]$ to actions.

**Semi-MDP** A Semi-Markov Decision Process (SMDP, Baykal-Gürsoy, 2010; Cinlar, 2013) is a generalization of the MDP formalism. It admits *temporally extended* actions, which, contrary to *primitive* ones (i.e., actions that execute for a single time step), can execute for a certain time during which the agent has no control over the decision process. A usual notion when treating SMDP is the *duration* or *holding time*, $\tau(s, a, h)$, which is the number of primitive steps taken inside a temporally extended action.

HRL builds upon the theory of Semi-MDPs, characterizing the concept of temporally extended action with basically two main formalisms (Pateria et al., 2021): sub-tasks (Dietterich, 2000) and options (Sutton et al., 1999). For the sake of this paper, we focus on the options framework.

**Options** An option (Sutton et al., 1999) is a possible formalization of a temporally extended action, which generalize the standard *primitive* ones. It can be seen as a fixed policy that acts on its own when selected until its termination. It consists of three components $o = (\mathcal{I}_o, \beta_o, \pi_o)$. $\mathcal{I}_o \subseteq \mathcal{S} \times [H]$ is the subset of states and stages pairs $(s, h)$ in which the option can start, $\beta_o : \mathcal{S} \times [H] \to [0, 1]$ defines the probability that an option terminates in a specific state-stage pair, $\pi_o : \mathcal{S} \times [H] \to \mathcal{A}$ is the policy executed until its termination. An example of an option could be a policy that, in a control problem, executes just a specific task, such as picking up an object.

**Problem Definition** Exactly as stated by Sutton et al. (1999, Theorem 1) *an MDP in which primitive actions $\mathcal{A}$ are replaced by options $\mathcal{O}$, becomes an SMDP.* This paper considers a hierarchical approach for problems with two-level of temporal abstractions. At the high level, the problem is seen as divided into different sub-tasks, the *options*, and at the low level, each sub-task is considered a problem on its own. More formally, on the top, the goal is to find the optimal policy $\mu : \mathcal{S} \times [H] \to \mathcal{O}$, which determines the optimal option for each state-instant pair, and thus defines the optimal sequence of sub-task to execute. On the other hand, at the bottom, the problem is to find the optimal option's policy to execute the respective sub-task individually. In this work, we first consider a situation in which the latter has already been solved, and a set of fixed policies are given. Then, we discuss a solution when this assumption fails. Regarding the former situation, an assumption is needed on the set of options (Fruit et al., 2017) to reach a proper optimal solution.

**Assumption 2.1** (Admissible options). The set of options $\mathcal{O}$ is assumed admissible, i.e. $\{\forall o \in \mathcal{O}, s \in \mathcal{S}, \text{ and } h \in [H], \exists o' \in \mathcal{O} : \beta_o(s, h) > 0 \text{ and } (s, h) \in \mathcal{I}_{o'}\}$.

**Average per-episode duration** In the following analysis, we will refer to $d$ as the average per-episode number of decisions taken in an episode of length $H$.

$$d = \frac{1}{K} \sum_{k=1}^{K} \sum_{o \in \mathcal{O}} \sum_{s \in \mathcal{S}} \sum_{h \in [H]} d_k(s, o, h) \tag{1}$$

where $d_k(s, o, h)$ is the number of times a temporally extended action (or option) $o$ has been selected in state $s$, in step $h$, in the episode $k$ of interaction with the environment. This quantity is a random variable, being dependent on the duration of the options, which is a random variable in turn.

## 2.1 Notation

In the following, we will use $\tilde{O}(\cdot)$ to indicate quantities that depend on $(\cdot)$ up to logarithmic terms. $\mathbb{1}(x = a)$ defines the indicator function

$$\mathbb{1}(x = a) \stackrel{def}{=} \begin{cases} 0, & \text{if } x \neq a \\ 1, & \text{if } x = a \end{cases}$$

In the analysis, we denote optimistic terms with $\sim$ and empirical ones with $\wedge$, e.g., $\tilde{p}$ *and* $\hat{r}$ *are, respectively, the optimistic transition model and the estimated reward function.*

## 3 Finite Horizon SMDP

In this section, we present a new formalism, *Finite-Horizon Semi-Markov Decision Processes*, that combines the notion used in FH-MDP with the concept of temporal abstraction.

A finite-horizon semi-MDP is defined as a tuple $\mathcal{SM} = (\mathcal{S}, \mathcal{O}, p, r, H)$, where $\mathcal{S}$ and $\mathcal{O}$ are the finite state and the temporally extended action spaces, respectively, $p(s', h'|s, o, h)$ is the probability of ending to state $s' \in \mathcal{S}$, after $(h' - h)$ steps, by playing the temporally extended action $o \in \mathcal{O}$ in state $s \in \mathcal{S}$ at stage $h \in [H]$. On the other hand, $r(s, o, h)$ is the expected reward accumulated until the termination of the temporally extended action $o \in \mathcal{O}$ played in state $s \in \mathcal{S}$ at stage $h \in [H]$ of the episode. Finally, $H$ is the horizon of interaction, and we define as $\tau(s, o, h)$ the random number of primitive steps taken when playing a temporally extended action $o$. The agent's behavior is modeled by a deterministic policy $\mu : \mathcal{S} \times [H] \to \mathcal{O}$ mapping a state $s \in \mathcal{S}$ and a stage $h \in [H]$ to a temporally extended action. The goal of the agent is to find a policy $\mu^*$ that maximizes the value function, defined as the expected sum of the rewards collected over the horizon of interaction and recursively defined as:

$$V^\mu(s, h) = \mathbb{E}_{(s', h') \sim p(\cdot|s, \mu(s, h), h)} \Big[ r(s, \mu(s, h), h) + V^\mu(s', h') \Big],$$

with the convention that $V^\mu(s, H) = 0$. The value function of any optimal policy is denoted by $V^*(s, h) := V^{\mu^*}(s, h)$.

To evaluate the performance of an algorithm in this setting, as commonly used in the provably efficient RL literature Auer et al. (2008); Azar et al. (2017); Zanette & Brunskill (2018); Fruit & Lazaric (2017), we define the regret in FH-SMDP as:

**Regret**    For any starting state $s \in \mathcal{S}$, and up to the episode $K$:

$$Regret(K) \stackrel{def}{=} \sum_{k=1}^{K} V^*(s, 1) - V^{\mu_k}(s, 1) \tag{2}$$

evaluates the performance of the policy learned until episode $k$, $V^{\mu_k}$ compared to the value of an optimal policy $V^*$, with $\mu^*$ the optimal policy selecting among options.

Even if this formalism is an extension of FH-MDP, it is impossible to apply the same algorithms used to solve FH-MDPs directly. The main complication, which will also appear in the analysis, is the uncertainty over the duration of the temporally extended actions. The number of decision steps taken in one episode is unknown because it depends on the random number of steps for which each selected temporally extended action executes.

For this reason, in the next section, we propose a novel algorithm for this setting.

## 4 FH-SMDP-UCRL

FH-SMDP-UCRL is a variant of the algorithm presented in Fruit & Lazaric (2017), which in turn is inspired by UCRL2 (Auer et al., 2008) and adapted for FH-SMDPs. This family of algorithms implements the principle of "*optimism in the face of uncertainty*", which states that when interacting in unknown environments—with an unknown model—the decisions have to be guided by a trade-off between what we believe is the best option and by a term representing the uncertainty on our estimates. More formally, the so-called *exploration bonus* is introduced, which quantifies the uncertainty level on our estimations of the model, computed from the observed samples. This exploration bonus is used to regulate the *exploration-exploitation* dilemma, inducing the algorithm to explore regions of the space with high uncertainty instead of sticking to what seems to be the optimal solution and to overcome situations in which the optimal solution resides in a region not yet discovered.

However, directly applying these algorithms in our setting is unfeasible, as they are designed for infinite-horizon average reward settings. Due to the lack of methods operating in these settings, we must design a new algorithm for finite-horizon SMDPs following the same paradigm.

---

**Algorithm 1:** UCRL-FH-SMDP

---

**Input:** $\mathcal{S}, \mathcal{O}$ with fixed policies, $H, K$

Initialize $\mu_0$ at random and $Q_1(s, o, h) = 0$ for all $(s, o, h) \in \mathcal{S} \times \mathcal{O} \times [H]$

Execute $\mu_0$ for $H$ steps and collect tuples $(s, o, h, s', h')$ and $r(s, o, h)$ to store in $\mathcal{D}_1$ ;

**for** $k = 1, \ldots, K$ **do**

    Compute $n_k(s, o, h)$;

    Estimate empirical SMDP $\widehat{\mathcal{SM}}_k = (\mathcal{S}, \mathcal{O}, \hat{p}_k, \hat{r}_k)$ with equation 3, equation 4.;

    Compute the confidence sets $B_k^r(s, o, h)$ and $B_k^p(s, o, h)$ using the confidence interval (equation 5, equation 6);

    Planning with Backward Induction for $\mu_k$, using an adaptation to finite horizon of *Extended Value Iteration* (Auer et al., 2008) (Algorithm 2);

    **for** $h = 1, \ldots, H$ **do**

        Execute $o = \mu_k(s, h)$ until it terminates;

        Observe $(s', h')$ and $r(s, o, h)$;

        Add the tuple $(s, o, h, s', h')_k$ and $r_k(s, o, h)$ to $\mathcal{D}_{k+1}$;

        Set $h = h'$;

    **end**

**end**

---

As displayed by Algorithm 1, at each episode $k$, we compute an estimate of the SMDP model, by computing, from the collected samples up to episode $k$, the empirical transition probability $\hat{p}_k$ and the reward function $\hat{r}_k$.

$$\hat{p}_k(s', h'|s, o, h) = \frac{\sum_{i=1}^{k-1} \mathbb{1}((s, o, h, s', h')_i = (s, o, h, s', h'))}{n_k(s, o, h)} \tag{3}$$

$$\hat{r}_k(s, o, h) = \frac{\sum_{i=1}^{k-1} r_i(s, o, h)}{n_k(s, o, h)} \tag{4}$$

We then redefine the confidence intervals of these two quantities, $\beta_k^p$ and $\beta_k^r$, respectively as

$$\beta_k^r(s, o, h) \;\propto\; \sqrt{\frac{2\hat{\mathrm{Var}}(r) \ln 2/\delta}{n_k(s, o, h)}} + \frac{7 \ln 2/\delta}{3(n_k(s, o, h-1)}, \tag{5}$$

$$\beta_k^p(s, o, h) \;\propto\; \sqrt{\frac{S \log\left(\frac{n_k(s, o, h)}{\delta}\right)}{n_k(s, o, h)}}, \tag{6}$$

where $\hat{\mathrm{Var}}(r)$ is the sample variance of the reward function. From the estimates and the confidence intervals just defined, we can build the confidence sets $B_k^p$ and $B_k^r$, which contain, with high probability, the true model. Then, we define $\mathcal{SM}_k$, the set of plausible SMDPs, characterized by rewards and transition models within the confidence sets. Given $\mathcal{SM}_k$ we can compute the optimistic policy $\tilde{\mu}_k$ and the relative optimistic value function $\tilde{V}^{\mu_k}$. We use an adaptation of *extended value iteration* (Auer et al., 2008) for FH-SMDP, Algorithm 2, which admits multi-step transitions and, thus, temporally extended actions.

Then, by playing this policy for an entire episode, we collect new samples and restart the process for the next episode $k + 1$.

## 4.1 Regret Analysis

In this section, we theoretically analyze the regret suffered by FH-SMDP-UCRL. In particular, we provide an upper bound highlighting the options set's dependency.

---

**Algorithm 2:** Extended Value Iteration for FH-SMDP

---

**Input:** $\mathcal{S}, \mathcal{O}, B_k^r, B_k^p$
Set $Q_{H+1}(s, o) = 0$ for all $(s, o) \in \mathcal{S} \times \mathcal{O}$;
**for** $h = H \dots 1$ **do**
    **for** $(s, o) \in \mathcal{S} \times \mathcal{O}$ **do**
        **for** $h' = h + 1 \dots H + 1$ **do**
            Compute;

$$Q_{hk}(s, o) = \max_{r \in B_k^r(s,o,h)} r(s, o, h) + \max_{p \in B_k^p(s,o,h)} \mathbb{E}_{s',h' \sim p(\cdot, \cdot | s, o, h)}[V_{h',k}(s')]$$

$$V_{h'k}(s) = \min\{H - (h' - 1), \max_{o \in \mathcal{O}} Q_{h'k}(s, o)\}$$

        **end**
    **end**
**end**
**Output:** $\mu_k(s, h) = \mathrm{argmax}_{o \in \mathcal{O}} Q_{hk}(s, o)$

---

**Theorem 4.1.** *Considering a non-stationary Finite Horizon SMDP $\mathcal{SM}$ and a set of options $\mathcal{O}$, with bounded primitive reward $r(s, a) \in [0, 1]$. The regret suffered by algorithm FH-SMDP-UCRL, in $K$ episodes of horizon $H$ is bounded as:*

$$Regret(K) \leq \tilde{O}\left(\left(\sqrt{SOKd^2}\right)\left(\overline{T} + \sqrt{S}H\right)\right)$$

*with probability $1 - \delta$.*
*Where:*

$$\overline{T} = \max_{s,o,h} \sqrt{\mathbb{E}[\tau(s, o, h)^2]}$$
$$= \max_{s,o,h} \sqrt{\mathbb{E}[\tau(s, o, h)]^2 + \mathrm{Var}[\tau(s, o, h)]},$$

*$\tau$ is the holding time, the random number of primitive steps taken when playing a temporally extended action $o$, and $d$ is the average per-episode number of played options.*

This result introduces one of the main contributions, an option-dependent upper bound on the regret in FH-MDP with options, not worst-case as in Fruit & Lazaric (2017). A dependency on the properties of an option set is introduced, embodied into both $\overline{T}$ and $d$. The former gives the same interesting consideration already underlined by Fruit & Lazaric (2017), whereby the extra cost of having actions with random duration is only partially additive rather than multiplicative. On the other hand, the latter emphasizes the real benefit of using a hierarchical approach over a flat one. Considering only the dominant term, the regret of FH-SMDP-UCRL scales with $\tilde{O}(HS\sqrt{OKd^2})$, where $d$ is the average per-episode number of played options, which directly depends on the average duration of each option selected. While the regret of UCRL2 adapted to the finite-horizon setting scales with $\tilde{O}(HS\sqrt{AKH^2})$ when considering non-stationary transitions. By definition, $d \leq H$, however, for specific classes of problems with a well-constructed hierarchy, $d \ll H$, and $O \leq A$, which finally proves the gain in the complexity of using a hierarchical approach over a standard one, clearly this is valid when a proper set of fixed options is given.

## 4.2 Derivation of FH-MDP and parallelism with AR-SMDP

To further strengthen the obtained result, we can show that considering some assumptions, we can derive the upper bound by Auer et al. (2008) adapted to FH-MDPs (Ghavamzadeh et al., 2020), and, by doing particular consideration, it is possible to underline some similarity between our result and the upper bound on the regret of the algorithm designed for average reward settings.

### 4.2.1 Finite Horizon MDP

Referring to the result provided by Auer et al. (2008) adapted to the finite horizon case (Ghavamzadeh et al., 2020), the regret in Finite Horizon MDPs scales with $\tilde{O}(HS\sqrt{AT})$, or with $\tilde{O}(H^{\frac{3}{2}}S\sqrt{AT})$ when the MDP is non-stationary, in the sense that the transition kernel is different for each step $h \in [H]$. If, in our upper bound, we substitute the cardinality of the option set $O$, with the primitive-action space $A$. This leads to having $\overline{T} = 1$ and $d = H$ because the primitive actions, by definition, terminate after a single time step. Thus, the average duration of these single-step options is 1, and the number of decisions taken in one episode is exactly $H$. Then, having bounded primitive reward $r(s,a) \in [0,1]$, we can write our result as $\tilde{O}(H^{\frac{3}{2}}S\sqrt{AKH})$ and considering the definition of $T = KH$ (Dann et al., 2017; Azar et al., 2017; Zanette & Brunskill, 2018), we obtain the same equation.

**Remark** We are aware of the tighter upper bounds in the Finite Horizon literature by Azar et al. (2017), that get rid of a $\sqrt{HS}$, by tightly bounding the estimation error $(\tilde{p} - p)\tilde{V}^{\mu_k}$ and their exploration bonus in terms of the variance of $V^*$ at the next state, and by using empirical Bernstein and Freedman's inequalities (Maurer & Pontil, 2009; Freedman, 1975). However, with this work, our primary focus is to emphasize the role played by the options set's composition instead of providing a very tight analysis. We still think the same tricks could be used in our analysis to tighten the bound, but we leave that for future work.

### 4.2.2 Parallelism with Average Reward Setting

Fruit & Lazaric (2017) showed that the regret paid by their algorithm, designed for the Average Reward setting when considering bounded holding times, and $R_{max} = 1$ is upper bounded by with $\tilde{O}(D_{\mathcal{O}}S_{\mathcal{O}}\sqrt{On} + T_{max}\sqrt{S_{\mathcal{O}}On})$. By doing the same, hence considering bounded holding times and $R_{max} = 1$, our result becomes of order $\tilde{O}(HS\sqrt{OKd^2} + T_{max}\sqrt{SOKd^2})$. It is clearly impossible to derive one setting from the other. Nevertheless, we can spot some similarities between the two bounds. For finite-horizon problems, the diameter $D$ coincides with the horizon $H$ (Ghavamzadeh et al., 2020). Besides, $Kd$ equals $n$, the number of decisions made up to episode $K$. Then, the state space $S$ is the state space of the SMDP in our formalism, which is the definition provided for $S_O$ in Fruit & Lazaric (2017). The only difference is the additional $\sqrt{d}$, which is considered in our work contrary to what has been done in Fruit & Lazaric (2017) due to non-stationary transitions.

Thus, we prove that our result is a generalization of the case of FH-MDP and closely relates to the result presented for the Average Reward Setting.

## 5 Renewal Process

In this section, we provide a more accurate analysis of $d$, the average per-episode number of options that are selected in an episode of Horizon $H$, which, as mentioned above, is a random variable. We resort to the renewal processes theory Smith (1958) to provide an upper bound on this quantity, which turns out to depend on both the horizon $H$ and the options set.

The number of options selected in one episode clearly depends on the random duration of each option; hence, it is a random variable, and we would like to bound it with some quantity. Resorting to the *Renewal Theory* (Smith, 1958), this has an analogy with the *Renewal Function* $m(t)$.

**Definition 5.1** (Renewal Process). Let $S_1, S_2 \ldots$ be a sequence of i.i.d. random variables with finite and non-zero mean, representing the random time elapsed between two consecutive events, defined as the holding time. For each $n > 0$ we define $J_n = \sum_{i=1}^{n} S_i$, as the time at which the $n^{th}$ event of the sequence terminates. Then, the sequence of random variables $X_t$, characterized as

$$X_t = \sum_{n}^{\infty} \mathbb{I}_{\{J_n \leq t\}} = \sup\{n : J_n \leq t\} \tag{7}$$

constitutes a Renewal Process $(X_t)_{t \geq 0}$, representing the number of consecutive events that occurred up to time $t$.

**Definition 5.2** (Renewal Function). Considering a renewal process $(X_t)_{t \geq 0}$, the renewal function $m(t)$ is the expected number of consecutive events that occurred by time $t$.

$$m(t) = \mathbb{E}[X_t]$$

Hence, it is possible to take inspiration from a bound of the renewal function to bound $d$ for a single episode of length $H$.

**Lemma 5.3.** *[Bound on number of options played in one episode] Considering a Finite Horizon SMDP $\mathcal{SM}$ with horizon $H$ and, $O$ options with duration $\tau_{min} \leq \tau \leq \tau_{max}$ and $\min_o(\mathbb{E}[\tau_o])$ the expected duration of the shorter option. The expected number of options played in one episode $\tilde{d}$ can be seen as the renewal function $m(t)$ of a renewal process up to the instant $H$. With probability $1 - \delta$, this quantity is bounded by*

$$\tilde{d} < \sqrt{\frac{32(\tau_{max} - \tau_{min})H(\ln 2 - \ln \delta)}{(\min_o(\mathbb{E}[\tau_o]))^3}} + \frac{H}{\min_o(\mathbb{E}[\tau_o])}$$

This bound can also be considered a bound for $d$, being worst-case, because the denominator is lower bounded by $\min_o(\mathbb{E}[\tau_o])$, which is the expected duration of the shorter option. Refer to the appendix for detailed proof of this result.

## 6 Option Learning

Clearly, having a set of pre-trained options does not properly characterizes a realistic scenario. It would be more interesting to study situations in which we relax the assumption of having a set of pre-trained options and consider known just their initial state set and termination conditions. We can highlight characteristics of problems more suited to be solved with a hierarchical approach rather than standard ones.

To address this analysis, we decide to present a model-based two-phase approach, Algorithm 4, which uses Algorithm 3 to learn each option policy individually and subsequently exploits them to solve the SMDP with FH-SMDP-UCRL. As Algorithm 3 shows, each option is considered as a single FH-MDP, defined based on its initial-state set and termination probability, as $\mathcal{M}_o = (\mathcal{S}_o, \mathcal{A}_o, p, r_o, H_o)$ where $\mathcal{S}_o \subseteq \mathcal{S}$, $\mathcal{A}_o \subseteq \mathcal{A}$, $H_o \leq H$, which means that each option operates on a restricted portion of the original problem, for a specific fixed horizon $H_o$. The option's optimal policy is the policy of the relative sub-FH-MDP computed until episode $K_o$, which is the number of episodes assigned to each option.

Nevertheless, if no assumption on the reward function is defined, the options' optimal policies could be sub-optimal regarding the optimal policy computed by a standard approach for that portion of the MDP, being the option's scope limited to a certain part of the MDP with the impossibility of having feedback on what happens after its termination.
Therefore we need to state:

**Assumption 6.1.** Given an MDP $\mathcal{M} = (\mathcal{S}, \mathcal{A}, p, r, H)$ and a set of options $o \in \mathcal{O}$, which define a Semi-MDP $\mathcal{SM}$. Define $\pi_o^*$ as the optimal policy of the option $o$ learned individually on the sub-MDP $\mathcal{M}_o = (\mathcal{S}_o, \mathcal{A}_o, p, r_o, H_o)$ with $\mathcal{S}_o \subseteq \mathcal{S}$, $\mathcal{A}_o \subseteq \mathcal{A}$, and $H_o \leq H$. The reward function $r_o$ of the sub-MDP $\mathcal{M}_o$, which could differ from $r$, ensure that

$$\pi^*(s) = \pi_o^*(s) \ \ \forall s \in \mathcal{S}_o \ , \forall o \in \mu^*(s)$$

with $\pi^*(s)$ the optimal policy on $\mathcal{M}$, and $\mu^*(s)$ the optimal policy on $\mathcal{SM}$.

This assumption guarantees that the computed option's optimal policy equals the optimal policy of the entire problem in the option's region for all the options selected by the optimal SMDP policy $\mu^*$.

This assumption is required to highlight the convenience of a hierarchical solution compared to a standard (flat) one. There exist several problems in which this condition is verified. For instance, let us consider a manipulation task in which a robot has to pick a die that shows a number, and place it in a differently defined location, changing the displayed number. This problem could be divided into four different individual

problems, the object pick-up, the rotation of the die, the movement of an object from a specific position to a second one, and the drop of the object. When considering all these problems individually, each optimal policy learned separately for each skill would equal an optimal policy learned when considering the problem as a whole for the same states. Indeed, the best possible policy to pick up an object would be similar if we consider the problem as a whole or restricted to a particular region of the state-action space, which is crucial just for this specific task.

With the result presented in the following section, this assumption characterizes class problems in which such an approach should be preferred to a standard one.

---

**Algorithm 3:** Option Learning

**Input:** the option $o$ and $K_o$

Compute $H_o$ for the option $o$;

Set $Q_{H_o+1}(s,a) = 0$ for all $(s,a) \in (\mathcal{S}_o \subseteq \mathcal{S}) \times (\mathcal{A}_o \subseteq \mathcal{A})$;

**for** $k = 1, \ldots, K_o$ **do**

    Sample randomly $(s,h)$ in $\mathcal{I}_o$;

    Compute $n_k(s,a,h)$;

    Estimate empirical MDP $\widehat{\mathcal{M}}_k = (\mathcal{S}_o, \mathcal{A}_o, \hat{p}_k, \hat{r}_k)$, with an adaptation of eq. 3 and 4 for the flat model.;

    Compute the confidence sets $B_k^r(s,a,h)$ and $B_k^p(s,a,h)$ using adaptation of the confidence interval (equation 5, equation 6) for the flat model;

    Planning by backward induction for $\pi_{o_{hk}}$ with Extended Value Iteration for FH-MDPs (Ghavamzadeh et al., 2020), in the horizon $H_o$.;

    Play $\pi_{o_k}$ for an episode to collect new samples.;

**end**

**Output:** $\pi_o^{K_o}$.

---

**Algorithm 4:** Two-Phase Algorithm

**Input:** set of options $\mathcal{O}$ with random policies, $K_o$, and $K$

**for** $o \in \mathcal{O}$ **do**

    $\pi_o^{K_o} = Option - Learning(o, K_o)$ ;

**end**

**Output:** $FH - SMDP - UCRL(\mathcal{S}, \mathcal{O}, H, K - K_o H_o)$

---

### 6.1 Theoretical Analysis Option Learning

Let's now analyze the regret suffered by the two-phase algorithm that first learns each option policy and then subsequently solves the relative SMDP. This result is then used to characterize the problems in question.

**Theorem 6.2.** *The regret paid by the two-phase learning algorithm until the episode $K$ is:*

$$Regret(K) \leq \tilde{O}\left(K^{\frac{2}{3}}\sqrt[3]{H_O^5 S_O^2 A_O O} + HS\sqrt{Od^2 K}\right)$$

*with $H_O = \max_{o \in \mathcal{O}} H_o$, and $S_O$, and $A_O$, respectively, the upper bounds on the cardinality of the state and action space of the sub-FH-MDPs.*

This result shows the regret paid by this two-phase approach for both option learning and the SMDP resolution. The second term, the non-dominant in the $K$ (number of episodes) dependency, is precisely the regret paid by FH-SMDP-UCRL. On the other hand, the dominant one comes from the two-phase analysis and encloses both the regret paid for the option learning and the bias paid for stopping this procedure at a certain point before the convergence to the optimal options policies. We note that scales with $K^{2/3}$, but it was expected for the Explore-Then-Commit nature of the proposed algorithm (Lattimore & Szepesvári, 2020). However, studying a more efficient algorithm is left for future work.

Given this result, it is now essential to understand if there are any situations in which such an approach could produce more benefit compared to a standard one. This would lead to defining classes of problems in which learning using a hierarchical approach almost from scratch should be preferred to a standard one.

In order to do so, we compare the regret paid by our method with the theoretical lower bound for flat FH-MDPs with non-stationary transition probabilities, which is $\Omega(H\sqrt{SAT})$ (Auer et al., 2008; Ghavamzadeh et al., 2020). If the comparison results in favor of the hierarchical approach, it would be theoretically possible to define the characteristics of the problem that would discriminate the preferred framework.

Let $\mathcal{R}$ be the ratio between the regret upper bound of our approach and the regret theoretical lower bound.

$$\mathcal{R} = \frac{Regret_{SMDP}}{\Omega(H\sqrt{SAT})} \leq \frac{K^{2/3}(H_O^5 S_O^2 A_O O)^{1/3}}{H^{3/2}\sqrt{SAK}} \tag{8}$$

By considering particular relations between the option-MDP and the original one, where $A_O = \alpha A$, $S_O = \alpha S$ and $H_O = \alpha H$, we can rewrite this ratio as:

$$\mathcal{R} \leq \frac{K^{1/6}\alpha^{8/3}H^{1/6}S^{1/6}O^{1/3}}{A^{1/6}} \tag{9}$$

$\mathcal{R} \leq 1$ is the condition that determines that our approach is preferable over the standard one. Thus, by imposing it, we can find the maximum number of episodes for which this constraint is satisfied.

$$K \leq \frac{A}{\alpha^{16}HSO^2} \tag{10}$$

The problems with characteristics compliant with this equation are the ones in which a hierarchical approach should be preferred to a standard one, even when no fixed sub-tasks policies (fixed *options*) are provided.

**Final Sub-Optimality Remark.** In both Theorem 4.1, 6.2, we need to consider that, by using a defined option set, we are introducing a bias. It could be that the optimal policy on the flat problem is irreproducible by a concatenation of the policies of the options chosen by the optimal high-level policy $\mu^*$. This is because the structure introduced by the options can cause some states to become inaccessible for the high-level SMDP. This issue is also treated by Fruit & Lazaric (2017) and produces an additional term on the regret equal to $V^*(M) - V^*(M_O)$, where $M$ is the primitive MDP and $M_O$ the same MDP with options.

## 7 Proofs sketch

In this section, we provide the sketch proofs of theorem 4.1 and theorem 6.2. Please refer to the appendix for all the details.

### 7.1 Sketch proof of theorem 4.1

We defined the regret in finite horizon problems as in eq. 2. Optimistic algorithms work by finding an optimistic estimation of the model of the environment to compute the optimistic value function and the optimistic policy. Considering how the confidence sets are constructed, we can state that $\tilde{p} \geq p$ and $\tilde{r} \geq r$, where terms without tilde are the real one, hence, $V^*(s,h) \leq \tilde{V}^{\mu_k}(s,h)$ for all $h$. Thus, we can bound eq. 2 with

$$Regret(K) \overset{opt}{\leq} \sum_{k=1}^{K} \tilde{V}^{\mu_k}(s,1) - V^{\mu_k}(s,1) \tag{11}$$

Let's now introduce a Performance Difference Lemma for FH-SMDPs.

**Lemma 7.1.** *[Performance Difference Lemma for FH-SMDP] Given two FH-SMDPs $\hat{M}$ and $\tilde{M}$ with horizon $H$, and respectively rewards $\hat{r}$, $\tilde{r}$ and transition probabilities $\hat{p}$, $\tilde{p}$. The difference in the performance of a policy $\mu_k$ is:*

$$\tilde{V}^{\mu_k}(s,1) - \hat{V}^{\mu_k}(s,1)$$

$$= \hat{\mathbb{E}}\Bigg[\sum_{i=1}^{H}\Big(\big(\tilde{r}(s_i,o_i,h_i) - \hat{r}(s_i,o_i,h_i)\big)$$

$$+ \big(\tilde{p}(s_{i+1},h_{i+1}|s_i,o_i,h_i) - \hat{p}(s_{i+1},h_{i+1}|s_i,o_i,h_i)\big)$$

$$\tilde{V}^{\mu_k}(s_{i+1},h_{i+1})\Big)\mathbb{1}\{h_i < H\}\Bigg]$$

*where $\hat{\mathbb{E}}$ is the expectation taken w.r.t. $\hat{p}$ and $\mu_k$.*

Note that the summation steps are not unitary but skip according to the length of the transitions $h' - h$. The derivation of this lemma follows the one provided by Dann et al. (2017) for FH-MDPS that is commonly used in literature (Azar et al., 2017; Zanette & Brunskill, 2018). Check the appendix for further details.

Now we can use lemma 7.1 to substitute the difference of value function in eq. 11 and we can upper bound both the difference of $r$ and $p$, with 2 times their confidence intervals and the optimistic value $\tilde{V}^{\mu_k}(s_{i+1},h_{i+1})$ with the horizon $H$ - we consider bounded primitive reward $r(s,a) \in [0,1]$.

$$Regret(K) \leq \sum_{k=1}^{K}\mathbb{E}\Bigg[\sum_{i=1}^{H}\Big(2\beta_k^r + 2\beta_k^p H\Big)\mathbb{1}\{h_i < H\}\Bigg]$$

In the Finite-Horizon literature(Dann et al., 2017; Zanette & Brunskill, 2018), two terms are commonly used in the proofs: (1) $w_k(s,o,h)$ that is the probability of taking the option $o$, in state $s$ at time step $h$, which depends on the policy $\mu_k$ and the transition probability of the real SMDP, (2) $L_k$, which defines the set of episodes visited sufficiently often, and the set of $(s,o,h)$ that were not visited often enough to cause high regret. Therefore, for using the same approach to conduct the proof, we can substitute the expectation $\mathbb{E}$, which is taken w.r.t. the policy $\mu_k$ and the real transition probability $p(s',h'|s,o,h)$, with

$$\sum_{(s,o,h)\in L_k} w_k(s,o,h)$$

We defined the confidence intervals of $r$ and $p$, as in the equations 5, 6, respectively using Empirical Bernstein Inequality (Maurer & Pontil, 2009), Hoeffding (1963) and Weissman et al. (2003).

By substituting these definitions and the term just introduced, we get, up to numerical constants, that the regret is bounded by

$$\sum_{k}\sum_{i\in[H]}\sum_{(s,o,h)\in L_k}\frac{w_k(s_i,o_i,h_i)}{\sqrt{n_k(s_i,o_i,h_i)}}\Bigg(\sqrt{\hat{\mathrm{Var}}(r)} + \sqrt{S}H + \frac{1}{\sqrt{n_k(s,o,h)}}\Bigg)$$

**Lemma 7.2.** *Considering a non-stationary MDP M with a set of options as an SMDP $M_{\mathcal{O}}$ (Sutton et al., 1999). In $M_{\mathcal{O}}$ the number of decisions taken in the $k^{th}$-episode is a random variable d and*

$$\sum_{i\in H}\sum_{(s,o)\in L_k} w_k(s_i,o_i,h_i)\mathbb{1}\{h_i < H\} = d \ with \ \{\forall k : d \leq H\}$$

*Therefore, the following holds true:*

$$\sum_{k}\sum_{i\in H}\sum_{(s,o)\in L_k} w_k(s_i,o_i,h_i)\sqrt{\frac{1}{n_k(s_i,o_i,h_i)}} = \tilde{O}\Big(\sqrt{SOKd^2}\Big)$$

Substituting the result of Lemma 7.2 in the equation of the regret, we get

$$Regret(K) \leq \tilde{O}\left(\left(\sqrt{SOKd^2}\right)\left(\sqrt{\hat{\mathrm{Var}}(r)} + \sqrt{S}H\right) + dSO\right) \tag{12}$$

where as mentioned above, $d$ is the expected number of decision steps taken in one episode, $n$ is the total number of decisions taken up to episode $k$, and $\hat{\mathrm{Var}}(r)$ is the empirical variance of the reward that emerged from the use of Empirical Bernstein inequality. A dependency on the variance of the reward is not that explainable for what we want to show; hence we upper bound this term by the square root of the empirical variance $\overline{T}$ of the duration of the options seen up to episode $k$, and this complete the proof.

**Challenge** The main challenge encountered during the analysis, which makes it different from the case of FH-MDPs, resides under the fact that in a Finite Horizon SMDP, the policy takes a number of temporally extended actions that are not defined, being the duration of a temporally extended action a random variable itself. Thus, it is not possible to recursively apply the standard bellman operator to define the performance difference lemma as in Dann et al. (2017) (Lemma E.5). More precisely, we have a summation of a number of elements that is a random variable, and thus it is unknown. This, and the fact that we resort to the renewal process theory (Smith, 1958), to upper bound this unknown variable ($d$, Section 5), describe the novelty in our analysis with respect to the standard analysis followed in the FH-MDP literature.

## 7.2 Sketch proof of theorem 6.2

In order to prove the regret paid by the two-phase algorithm, we first consider that we can write the regret as the sum of the regret paid in the first phase and the regret paid in the second one, plus an additional bias term. In the first phase, we pay full regret for each option learning. Then, the bias is defined as the maximum average regret considering the option learning as a finite horizon MDP with horizon $H_o$. Finally, the last term equals the regret of the SMDP learning with fixed options for the remaining episodes.

$$Regret(K) \leq \underbrace{\sum_{o \in O} K_o H_o}_{option\ learning} + \underbrace{K_2 \max_{o \in O} \frac{1}{K_o} H_o^2 S_o \sqrt{A_o K_o}}_{bias} + \underbrace{HS\sqrt{Od^2 K_2}}_{FH-SMDP-UCRL\ regret} \tag{13}$$

Where $K_2$ are the episodes used for the SMDP learning and $K = \sum_{o \in O} K_o + K_2$. Then considering that we allocate $K_o$ episodes for each option learning and upper bound $K_2 \leq K$, by bounding $A_O$, $S_O$, and $H_O$ respectively, with the upper bounds on the action-space cardinality, state-space cardinality, and horizon of the options set, we can get rid of the $\max_{o \in \mathcal{O}}$, and solve the equation in closed form to find the $K_o$ which minimizes the regret.

$$K_o = \sqrt[3]{\frac{K^2 S_O^2 H_O^2 A_O}{O^2 4}} \tag{14}$$

Now, we conclude the proof by substituting equation 14 in equation 13.

## 8 Related Works

In the FH-MDP literature, several works analyze the regret of different algorithms. Osband & Van Roy (2016) present a lower bound that scales with $\Omega(\sqrt{HSAT})$. On the other hand, many other works propose various upper bounds for their algorithm. The most common upper bound is the adaptation of Auer et al. (2008) proposed by Ghavamzadeh et al. (2020), which is of the order of $O(HS\sqrt{AT})$. This result has then been improved in the following papers. An example is Azar et al. (2017), which proposes a method with an upper bound on the regret of $O(\sqrt{HSAT})$ that successfully matches the lower bound.

Nevertheless, only some works focused on theoretically understanding the benefits of hierarchical reinforcement learning approaches, and, to the best of our knowledge, this is the first to analyze these aspects in FH-SMDPs. To conduct our analysis, we take inspiration from the paper by Fruit & Lazaric (2017), in

which they propose an adaptation of UCRL2 (Auer et al., 2008) for SMDPs. They first study the regret of the algorithm for general SMDPs and then focus on the case of MDP with options, providing both a lower bound and a worst-case upper bound. This work was the first that theoretically compares the use of options instead of primitive actions to learn in SMDPs. Nonetheless, it focuses on the average reward setting to study how it is possible to induce a more efficient exploration by using options, and it assumes fixed options. Differently, we aim to analyze the advantages of using options to reduce the sample complexity of the problem, resorting to the intuition that temporally extended actions can intrinsically reduce the planning horizon in FH-SMDPs. Furthermore, we provide an *option-dependent* upper bound, instead of a worst-case one, that better quantifies the impact of the option duration on the regret. Other works providing a theoretical analysis of hierarchical reinforcement learning approaches are Fruit et al. (2017), which is an extension of the previous work in which the need for prior knowledge of the distribution of cumulative reward and duration of each option is relaxed. Even in this case, they consider the average reward setting, and the objective is identical.

Then, Mann et al. (2015) study the convergence property of Fitted Value Iteration (FVI) using temporally extended actions, showing that a longer options duration and pessimistic value function estimates lead to faster convergence. Finally, Wen et al. (2020) demonstrate how patterns and substructures in the MDP benefit planning speed and statistical efficiency. They present a Bayesian approach exploiting this information and analyze how sub-structure similarities and sub-problems' complexity contribute to the regret of their algorithm.

## 9 Conclusions

In conclusion, we propose a new algorithm for Finite Horizon Semi Markov decision processes called FH-SMDP-UCRL, and we provide theoretical evidence that supports our original claim. Using hierarchical reinforcement learning, it is provably possible to reduce the problem complexity of a Finite Horizon problem when using a well-defined set of options. This analysis is the first for FH-SMDP and provides a form of option-dependent analysis for the regret that could be used to define objectives for options discovery methods better. Furthermore, by relaxing the assumption of having a set of fixed options policies, we were able to provide insights on classes of problems in which a hierarchical approach from scratch would still be beneficial compared to a flat one. In the future, it could be interesting to analyze the HRL problem by resorting to the framework proposed in Foster et al. (2021), and the associated multi-purpose method, to provide an alternative to our approach. Moreover, we would like to improve the algorithm proposed for options learning to tighten the theoretical guarantees and further characterize this family of problems. Finally, following the ideas of Wen et al. (2020), we would like to investigate how the structure of the MDP could appear in our bound, which, in our opinion, is a fundamental point to put another brick in the direction of total understanding on the promising power of HRL.

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

# A   Detailed proof of Theorem 4.1

## A.1   Notation and Setting

First, we need to define the value function of an SMDP. In Sutton et al. (1999) it is defined as a formalism for MDP with options, that itself, by the demonstration presented in the same article, is an SMDP.
In our case, however, for the SMDP model, we are considering an additional dependency on $h \in [0, H]$.
Notation used:

- $H$ is the horizon

- $\mu$ policy over options $\{\mu : S \times O \times H \to [0, 1]\}$

- $r(s, o, h)$ is the discounted cumulative reward gained by selecting the option $o$, in state $s$, in the instant $h$ of the horizon $H$

- $p(s', h'|s, o, h)$ is a new transition model that characterizes both the state dynamic and the time the option executes.

- $w(s, o, h)$ is the probability of playing option $o$ being in state $s$ at time-step $h$

The value function is defined as:

$$V^\mu(s, h) = \sum_{o \in O_s} \mu(s, o, h) \Big[ r(s, o, h) + \sum_{s', h' > h} p(h', s'|s, o, h) V^\mu(s', h') \Big] \tag{15}$$

with $V^\mu(s, H) = 0$.

## A.2   Performance Difference Lemma

**Lemma 7.1.** *[Performance Difference Lemma for FH-SMDP] Given two FH-SMDPs $\hat{M}$ and $\tilde{M}$ with horizon $H$, and respectively rewards $\hat{r}$, $\tilde{r}$ and transition probabilities $\hat{p}$, $\tilde{p}$. The difference in the performance of a policy $\mu_k$ is:*

$$\tilde{V}^{\mu_k}(s, 1) - \hat{V}^{\mu_k}(s, 1)$$

$$= \hat{\mathbb{E}} \Big[ \sum_{i=1}^{H} \Big( \big( \tilde{r}(s_i, o_i, h_i) - \hat{r}(s_i, o_i, h_i) \big)$$

$$+ \big( \tilde{p}(s_{i+1}, h_{i+1}|s_i, o_i, h_i) - \hat{p}(s_{i+1}, h_{i+1}|s_i, o_i, h_i) \big)$$

$$\tilde{V}^{\mu_k}(s_{i+1}, h_{i+1}) \Big) \mathbb{1}\{h_i < H\} \Big]$$

*where $\hat{\mathbb{E}}$ is the expectation taken w.r.t. $\hat{p}$ and $\mu_k$.*

*Proof.* $\hat{\mathbb{E}}$ is the expectation taken w.r.t. the policy $\mu$ and the transition probability $\hat{p}(s', h'|s, o, h)$, and can be rewrite as:

$$\prod_{i=1}^{H} \mu_k(s_i, o_i, h_i) \hat{p}_k(s_{i+1}, h_{i+1}|s_i, o_i, h_i) \mathbb{1}\{h_i < H\}$$

This quantity is the distribution of visits for the policy $\mu_k$ in the "$\,\hat{}\,$" SMDP and it is equivalent to $w_{tk}$ for the FH-MDP case. The result follows by unrolling equation 15. Lemma E.5 Dann et al. (2017) for an example in FH-MDPs. $\qquad\square$

### A.3 Confidence Intervals

The confidence sets are defined as:

$$B_k^r(s,o,h) := [\hat{r}_k(s,o,h) - \beta_k^r(s,o,h),\ \hat{r}_k(s,o,h) + \beta_k^r(s,o,h)]$$
$$B_k^p(s,o,h) := \left\{ p_k(\cdot,\cdot|s,o,h) \in \nabla(s) : \|\tilde{p}_k(\cdot,\cdot|s,o,h) - \hat{p}_k(\cdot,\cdot|s,o,h)\|_1 \leq \beta_k^p(s,o,h) \right\}$$

and the relative confidence bounds $\beta_k^r(s,o,h)$ and $\beta_k^p(s,o,h)$ using Empirical Bernstein bound (Maurer & Pontil, 2009), Hoeffding (1963) and Weissman et al. (2003).

$$\beta_k^r(s,o,h) \ \propto \ \sqrt{\frac{2\hat{\text{Var}}(r)\ln 2/\delta}{n(s,o,h)}} + \frac{7\ln 2/\delta}{3(n-1)} \tag{16}$$

$$\beta_k^p(s,o,h) \ \propto \ \sqrt{\frac{S\log\left(\frac{n_k(s,o,h)}{\delta}\right)}{n_k(s,o,h)}} \tag{17}$$

with $\hat{\text{Var}}(r)$ be the sample variance of r.

$$\hat{\text{Var}}(r) = \frac{1}{n(n-1)} \sum_{1 \leq i \leq j \leq n} (r_i - r_j)^2 \tag{18}$$

### A.4 Actual Proof

**Theorem 4.1.** *Considering a non-stationary Finite Horizon SMDP $\mathcal{SM}$ and a set of options $\mathcal{O}$, with bounded primitive reward $r(s,a) \in [0,1]$. The regret suffered by algorithm FH-SMDP-UCRL, in $K$ episodes of horizon $H$ is bounded as:*

$$Regret(K) \leq \tilde{O}\left( \left(\sqrt{SOKd^2}\right)\left(\overline{T} + \sqrt{S}H\right) \right)$$

*with probability $1 - \delta$.*
*Where:*

$$\overline{T} = \max_{s,o,h} \sqrt{\mathbb{E}[\tau(s,o,h)^2]}$$
$$= \max_{s,o,h} \sqrt{\mathbb{E}[\tau(s,o,h)]^2 + \text{Var}[\tau(s,o,h)]},$$

*$\tau$ is the holding time, the random number of primitive steps taken when playing a temporally extended action $o$, and $d$ is the average per-episode number of played options.*

*Proof.*

$$Regret(K) = \sum_{k=1}^{K} V^*(s,1) - \bar{V}^{\mu_k}(s,1)$$

$$\overset{Opt.}{\leq} \sum_{k=1}^{K} \tilde{V}^{\mu_k}(s,1) - \bar{V}^{\mu_k}(s,1)$$

$$= \sum_{k=1}^{K} \bar{\mathbb{E}}\left[ \sum_{i=1}^{H} \left( \left(\tilde{r}(s_i,o_i,h_i) - \bar{r}(s_i,o_i,h_i)\right) + \left(\tilde{p}(s_{i+1},h_{i+1}|s_i,o_i,h_i) - \bar{p}(s_{i+1},h_{i+1}|s_i,o_i,h_i)\right)\tilde{V}^{\mu_k}(s_{i+1},h_{i+1}) \right) \right.$$

$$\mathbb{1}\{h_i < H\} \bigg]$$

$$\overset{a}{=} \sum_k \sum_{i\in[H]} \sum_{(s,o,h)\in L_k} w_k(s_i, o_i, h_i)\bigg( \big(\tilde{r}(s_i, o_i, h_i) - \bar{r}(s_i, o_i, h_i)\big)$$

$$+ \big(\tilde{p}(s_{i+1}, h_{i+1}|s_i, o_i, h_i) - \bar{p}(s_{i+1}, h_{i+1}|s_i, o_i, h_i)\big)^T \tilde{V}^{\mu_k}(s_{i+1}, h_{i+1})\bigg)$$

$$\overset{b}{\le} \sum_k \sum_{i\in[H]} \sum_{(s,o,h)\in L_k} w_k(s_i, o_i, h_i)\Big(2\beta_k^r(s_i, o_i, h_i) + 2\beta_k^p(s_i, o_i, h_i)^T H\Big)$$

$$\overset{c}{\propto} \sum_k \sum_{i\in[H]} \sum_{(s,o,h)\in L_k} w_k(s_i, o_i, h_i)\bigg(\sqrt{\frac{\hat{\mathrm{Var}}(r)}{n_k(s, o, h)}} + \frac{1}{n_k(s, o, h) - 1} + \sqrt{\frac{S}{n_k(s, o, h)}}H\bigg)$$

$$\overset{d}{\le} \sum_k \sum_{i\in[H]} \sum_{(s,o,h)\in L_k} \frac{w_k(s_i, o_i, h_i)}{\sqrt{n_k(s_i, o_i, h_i)}}\Big(\sqrt{\hat{\mathrm{Var}}(r)} + \sqrt{S}H\Big) + \sum_k \sum_{i\in[H]} \sum_{(s,o,h)\notin L_k} \frac{w_k(s_i, o_i, h_i)}{n_k(s, o, h)}$$

$$\overset{e}{\le} \tilde{O}\bigg(\big(\sqrt{dSOn}\big)\Big(\sqrt{\hat{\mathrm{Var}}(r)} + \sqrt{S}H\Big) + dSO\bigg)$$

$$\overset{f}{\le} \tilde{O}\bigg(\big(\sqrt{dSOn}\big)\Big(R_{max}\overline{T} + \sqrt{S}H\Big) + dSO\bigg)$$

with

$$\overline{T} = \max_{s,o,h} \sqrt{\mathbb{E}[\tau(o_i, s_i, h_i)^2]} = \max_{s,o,h} \sqrt{\mathbb{E}[\tau(o_i, s_i, h_i)]^2 + \mathrm{Var}[\tau(s_i, o_i, h_i)]} \tag{19}$$

with $\tau$ representing the average duration of the set of options seen so far.

The first passage is a standard inequality when proving the regret in frameworks adopting optimism in face of uncertainty.

(a) The expectation with respect to the policy $\mu_k$ and the transition model $\bar{p}$ can be replaced with a more common formulation used in the Finite Horizon literature (Dann et al., 2017; Zanette & Brunskill, 2018), $\sum_{(s,o,h)\in L_k}$.
Where, $L_k$ is defined as the good set (Dann et al., 2017; Zanette & Brunskill, 2018), which is the number of episodes in which the triple $(s, o, h)$ is seen sufficiently often, and this equation is valid for all the tuples $(s, o, h)$ being part of this set.

(b) We upper bound the difference of rewards and transition probabilities with two times their relative confidence intervals, and, the Value function at the next step with the horizon length $H$.

(c) We substitute the confidence intervals with their definitions (eq. 17) neglecting logarithmic terms.

(d) We divide the summation in two, to upper bound the terms separately

(e) Using the adaptation of lemma 16 of Zanette & Brunskill (2018) for SMDPs, lemma 7.2, for the first term. Using passage (b) and (c) in the proof of lemma E.1

(f) Upperbounding the sample variance of $r$, with $R_{max}\overline{T}$. Where $\overline{T}$ is the sample variance of the duration.

$\square$

# B   Special Case of fixed-length options

Let's consider the same finite horizon MDP with options with fixed length $M :=< S, O, R_h, P_h, H, \bar{\tau} >$ where each option $o :=< I, \pi^o, \beta >$ has a fixed initial set $I$ and fixed termination condition $\beta$, $R_h(s,o) \in [0, \bar{\tau} R_{max}]$ is the expectation of the reward function distribution, $P_h(\cdot|s,o)$ is the transition distribution, $H$ is the horizon, and $\bar{\tau} \leq H$ is the options fixed length. The solution of the MDP will be a policy $\pi^H : S \rightarrow O$ that maximizes the cumulative return choosing among options' optimal policies $\pi^{o_i}$. The reward function over $state - options$ pairs relates to the flat-MDP's reward as:

$$R_h(s,o) = \mathbb{E}_{\substack{s_0=s \\ a_i \sim \pi^o(\cdot|s_i) \\ s_{i+1} \sim p_{h+1}(\cdot|s_i,a_i)}} \left[ \sum_{i=0}^{\bar{\tau}-1} r_{h+i}(a_i, s_i) \right]$$

Denote with $V_n^{\pi^H}(s)$ the state value function associated with a hierarchical policy $\pi^H$ (with Hierarchical policy we define a policy that chooses among options).

$$V_n^{\pi^H}(s) = \mathbb{E}_{\substack{s_0=s \\ o_j \sim \pi^H(\cdot|s_j) \\ s_{j+1} \sim P(\cdot|s_j,o_j)_h}} \left[ \sum_{j=0}^{N} R_{h \times \bar{\tau}}(s_j, o_j) \right]$$

with $N = \frac{H}{\bar{\tau}}$ the number of decision steps that occurs during the Horizon.

In this way, we can exploit the same *performance difference lemma* of Dann (Dann et al., 2017) Lemma E.15, where, instead of actions we have fixed length options, and we sum over $N$ decision steps. Hence, we can write:

$$Regret(K) \overset{Opt.}{\leq} \sum_{k=1}^{K} \tilde{V}_1^{\pi_k^H}(s) - \bar{V}_1^{\pi_k^H}(s) \tag{20}$$

$$\overset{a}{=} \sum_k \sum_{n \in [N]} \sum_{(s,o) \in L_k} w_{nk}(s,o) \left( \left( \tilde{R}_n(s,o) - \bar{R}_n(s,o) \right) + \left( \tilde{P}_n(s,o) - \bar{P}_n(s,o) \right)^T \tilde{V}_{n+1}^{\pi_k} \right) \tag{21}$$

$$+ \; term \; considering \; the \; state\text{-}options \; pairs \; inside \; the \; failure \; event \tag{22}$$

here $w_{nk}(s,o)$ is the probability of visiting state $s$ and choosing option $o$ there at the decision step n in the $k$-th episode.

Then, we consider a new formulation of the confidence sets:

$$B_{nk}^R(s,o) := [\hat{R}_{nk}(s,o) - \beta_{nk}^R(s,o), \; \hat{R}_{nk}(s,o) + \beta_{nk}^R(s,o)]$$
$$B_{nk}^P(s,o) := \left\{ P_{nk}(\cdot|s,o) \in \nabla(s) : \|\tilde{P}_{nk}(\cdot|s,o) - \hat{P}_{nk}(\cdot|s,o)\|_1 \leq \beta_{nk}^P(s,o) \right\}$$

using Hoeffding (1963) and Weissman et al. (2003) the confidence bounds are:

$$\beta_{nk}^R(s,o) \; \propto \; R_{max}\bar{\tau}\sqrt{\frac{\log\left(\frac{n_{nk}(s,o)}{\delta}\right)}{n_{nk}(s,o)}} \tag{23}$$

$$\beta_{nk}^P(s,o) \; \propto \; \sqrt{\frac{S\log\left(\frac{n_{nk}(s,o)}{\delta}\right)}{n_{nk}(s,o)}} \tag{24}$$

After the definition of the confidence sets we can bound the previous equation as follow:

$$\sum_{k=1}^{K} \tilde{V}_1^{\pi_k^H}(s) - \bar{V}_1^{\pi_k^H}(s) = \sum_k \sum_{n\in[N]} \sum_{(s,o)\in L_k} w_{nk}(s,o)\Big( \big(\tilde{R}_{nk}(s,o) - \bar{R}_{nk}(s,o)\big) + \big(\tilde{P}_{nk}(s,o) - \bar{P}_{nk}(s,o)\big)^T \tilde{V}_{h+1}^{\pi_k} \Big)$$

$$+ \text{ term considering the state-options pairs inside the failure event}$$

$$\overset{a}{\leq} \sum_k \sum_{n\in[N]} \sum_{(s,o)\in L_k} w_{nk}(s,o)\Big( 2\beta_{nk}^R + 2\beta_{nk}^{P\,T}(H-\bar{\tau}) \Big)$$

$$\overset{b}{\propto} \sum_k \sum_{n\in[N]} \sum_{(s,o)\in L_k} w_{nk}(s,o)\Big( \frac{R_{max}\bar{\tau}}{\sqrt{n_{nk}}} + \sqrt{\frac{S}{n_{nk}}}(H-\bar{\tau}) \Big)$$

$$= \sum_k \sum_{n\in[N]} \sum_{(s,o)\in L_k} \frac{w_{nk}(s,o)}{\sqrt{n_{nk}}}\Big( R_{max}\bar{\tau} + \sqrt{S}(H-\bar{\tau}) \Big)$$

$$\overset{c}{\leq} \tilde{O}\Big( N\sqrt{SOK}\big(R_{max}\bar{\tau} + \sqrt{S}H - \sqrt{S}\bar{\tau}\big) \Big)$$

$$\overset{d}{\leq} \tilde{O}\Big( N\sqrt{SOK}\big(R_{max}\bar{\tau} + \sqrt{S}H\big) \Big)$$

(a) substituting the $\tilde{R}_n(s,o) - \bar{R}_n(s,o)$ and $\tilde{P}_n(s,o) - \bar{P}_n(s,o)$ with double the relative confidence interval and considering $\tilde{V}_{h+1}^{\pi_k} \leq (H - \bar{\tau})$. The second term will be omitted for ease of notation.

(b) replacing the confidence intervals with their definition

(c) Lemma E.2

(d) considering the worst case, where there isn't the negative term

comparing it with the bound of Fruit & Lazaric (2017) for bounded holding time:

$$\tilde{O}\Big( \big(D_o\sqrt{S} + T_{max} + (T_{max} - T_{min})\big)R_{max}\sqrt{S_O On} \Big)$$

that having options with fixed duration $\bar{\tau}$, and considering $R_{max} = 1$ reduces to:

$$\tilde{O}\Big( DS\sqrt{On} + \bar{\tau}\sqrt{SOn} \Big)$$

we have the same bound where instead of the diameter we have the Horizon $H$, and where $NK$ is exactly equal to the number of the decisions up to episode $k$, which is $n$ in their notation. We have:

$$\tilde{O}\Big( HS\sqrt{ON^2K} + \bar{\tau}\sqrt{SON^2K} \Big)$$

**Important**: Note that we have an additional $\sqrt{N}$ terms because we considered non-stationary MDP. This is a well-known penalty term when considering non-stationarity in the process.

## C    Proof of Theorem 6.2

**Theorem 6.2.** *The regret paid by the two-phase learning algorithm until the episode $K$ is:*

$$Regret(K) \leq \tilde{O}\Big( K^{\frac{2}{3}}\sqrt[3]{H_O^5 S_O^2 A_O O} + HS\sqrt{Od^2K} \Big)$$

*with $H_O = \max_{o\in\mathcal{O}} H_o$, and $S_O$, and $A_O$, respectively, the upper bounds on the cardinality of the state and action space of the sub-FH-MDPs.*

*Proof.* The regret of the two-phase algorithm can be written in this form

$$Regret(K) = \sum_{k=1}^{K_1} V_*^*(s,1) - V_{(\pi_k)}^\mu(s,1) + \sum_{k=k_1} V_*^*(s,1) - V_{\pi_{K_1}}^{\mu_k}$$

$$= \underbrace{\sum_{k=1}^{K_1} V_*^*(s,1) - V_{(\pi_k)}^\mu(s,1)}_{\text{Options learning Regret}} + \sum_{k=K_1}^{K} \underbrace{V_*^*(s,1) - V_{(\pi_{K_1})}^*}_{\text{Bias}} + \underbrace{V_{(\pi_{K_1})}^* - V_{(\pi_{K_1})}^{\mu_k}}_{\text{Regret SMDP with fixed options}}$$

The regret is the sum of the regret paid in the first phase and the regret paid in the second one, plus an additional bias term.

By assuming to find the option's policies by solving $O$ different sub-FH-MDPs, and giving all options equal samples $K_o$ ($K_1 = \sum_{o \in O} K_o$), we can write the regret paid in $K = \sum_{o \in O} K_o + K_2$ episodes as

$$Regret(K) \leq \sum_{o \in O} K_o H_o + K_2 \max_{o \in O} \frac{1}{K_o} H_o^2 S_o \sqrt{A_o K_o} + HS\sqrt{Od^2 K_2}$$

where $H_o$ is each sub-FH-MDP horizon, and $A_o$ and $S_o$ are the cardinality of the action and state spaces.

We consider paying full regret for each option learning, then the maximum average regret considering the option learning as a finite horizon MDP with horizon $H_o$, and the regret of the SMDP learning with fixed options. However, we can further simplify the regret if we assume $S_O, A_O, H_O$ to be the upper bounds of the relative quantities, getting rid of the maximization in the second term, and the regret became

$$Regret(K) \leq OK_o H_O + \frac{K_2}{K_o} H_O^2 S_O \sqrt{A_O K_o} + HS\sqrt{OK_2 d^2}$$

Now, by substituting $K_2$ and upper bounding $(K - OK_o) \leq K$ we can solve in closed form to find $K_o$ to minimize the regret.

$$Regret(K) \leq OK_o H_O + \frac{K}{K_o} H_O^2 S_O \sqrt{A_O K_o} + HS\sqrt{OK d^2}$$

$$K_o = \sqrt[3]{\frac{K^2 S_O^2 H_O^2 A_O}{O^2 4}}$$

Therefore, by substituting $K_o$ in the original equation, we have

$$Regret(K) \leq \tilde{O}\left( K^{\frac{2}{3}} (H_O^5 S_O^2 A_O O)^{\frac{1}{3}} + HS\sqrt{OK d^2} \right)$$

$\square$

Now we can compare the regret of this algorithm compared to the regret of UCRL2 adapted for non-stationary FH-MDPs (Ghavamzadeh et al., 2020).

$$Regret(UCRL2 - CH) \leq \tilde{O}(H^2 S\sqrt{AK})$$

$$\frac{Regret_{SMDP}}{Regret_{MDP}} \leq \frac{K^{1/6}\alpha^{8/3}O^{1/3}}{(HS)^{1/3}A^{1/6}} \leq 1 \tag{25}$$

$$K \leq \frac{H^2 S^2 A}{\alpha^{16} O^2} \tag{26}$$

## D    Renewal Processes

**Lemma D.1** (Renewal Function Bound). *Considering a Renewal process, $(X_t)_{t\geq 0}$, and a sequence $S_1, S_2 \dots$ of random variables, characterizing the random duration of an event, alternatively defined as holding time, with $supp(S_i) \in \{1, \dots, H\}$. We can bound, with probability $1 - \delta$, the expected number of random events that occurred up to time $t$, $X_t$, with:*

$$X_t < \sqrt{\frac{\ln 2 - \ln \delta}{cK}} + \frac{t}{\mu}$$

*with $c = \frac{\mu^3}{32\sigma^2 T}$ where $\mu$ is the mean of the r.v.s and $\sigma^2$ the variance.*

*Proof.* Based on the proof presented on Pinelis (2019), which apply DKW type inequalities to renewal processes (Dvoretzky et al., 1956)

$$\Pr\left(\sup_{0 \leq t \leq T} \left| \frac{X_{nt}}{n} - \frac{t}{\mu} \right| \geq \epsilon \right) \leq 2e^{-cn\epsilon^2}$$

Now we can equal $2e^{-cn\epsilon^2}$ to $\delta$ and find $\epsilon$.

$$\epsilon = \sqrt{\frac{\ln 2 - \ln \delta}{cn}}$$

Thus with probability $1 - \delta$

$$X_t \leq \sqrt{\frac{\ln 2 - \ln \delta}{cn}} + \frac{t}{\mu}$$

that completes the proof $\qquad\square$

**Lemma 5.3.** *[Bound on number of options played in one episode] Considering a Finite Horizon SMDP $\mathcal{SM}$ with horizon $H$ and, $O$ options with duration $\tau_{min} \leq \tau \leq \tau_{max}$ and $\min_o(\mathbb{E}[\tau_o])$ the expected duration of the shorter option. The expected number of options played in one episode $\tilde{d}$ can be seen as the renewal function $m(t)$ of a renewal process up to the instant $H$. With probability $1 - \delta$, this quantity is bounded by*

$$\tilde{d} < \sqrt{\frac{32(\tau_{max} - \tau_{min})H(\ln 2 - \ln \delta)}{(\min_o(\mathbb{E}[\tau_o]))^3}} + \frac{H}{\min_o(\mathbb{E}[\tau_o])}$$

*Proof.* The proof followed the one of lemma D.1 and the fact that we are considering $T = H$, $n = 1$, $t = H$, $\mu = \bar{\tau}$, $\sigma^2 = (\tau_{max} - \tau_{min})$, and $X_t = \tilde{d}$. Moreover, considering $\bar{\tau}$, the average holding time of the option, we can further state that $\bar{\tau} > min_o(E[\tau_o])$, which is the expected duration of the shorter option, and complete the proof. $\qquad\square$

## E    Useful Lemmas

**Lemma E.1** (lemma 16 (Zanette & Brunskill, 2018) for non stationary MDPs). *The following holds true:*

$$\sum_k \sum_{h \in [H]} \sum_{(s,a) \in L_k} w_{hk}(s,a) \sqrt{\frac{1}{n_k(s,a,h)}} = \tilde{O}(\sqrt{HSAT})$$

*where the extra $\sqrt{H}$ is due to the non-stationarity of the environment*

*Proof.*

$$\sum_{k} \sum_{h \in [H]} \sum_{(s,a) \in L_k} w_{hk}(s,a) \sqrt{\frac{1}{n_k(s,a,h)}}$$

$$\overset{a}{\leq} \sqrt{\sum_{k} \sum_{h \in [H]} \sum_{(s,a) \in L_k} w_{hk}(s,a)} \sqrt{\sum_{k} \sum_{h \in [H]} \sum_{(s,a) \in L_k} w_{hk}(s,a) \frac{1}{n_k(s,a,h)}}$$

$$\overset{b}{=} \sqrt{KH} \sqrt{\sum_{k} \sum_{h \in [H]} \sum_{(s,a) \in L_k} w_{hk}(s,a) \frac{1}{n_k(s,a,h)}}$$

$$\overset{e}{\leq} \tilde{O}(\sqrt{HSAT})$$

Then:

$$\sum_{k} \sum_{h \in [H]} \sum_{(s,a) \in L_k} \frac{w_{hk}(s,a)}{n_k(s,a,h)}$$

$$\overset{c}{\leq} \sum_{h \in [H]} \sum_{(s,a) \in L_k} \sum_{k} \frac{w_{hk}(s,a)}{\frac{1}{4} \sum_{j \leq k} w_{hj}(s,a)}$$

$$\overset{d}{\leq} 4HSA \log\left(\frac{Ke}{w_{min}}\right)$$

$$\overset{\sim}{\propto} HSA$$

(a) by Cauchy-Schwartz

(b) $\sum_{t \in [H]} \sum_{(s,a) \in L_k} w_{tk}(s,a) = H$ lemma 17 (*b*) Zanette & Brunskill (2018)

(c) lemma 2 Zanette & Brunskill (2018) adapted to the non-stationary case

(d) lemma E.5 Dann et al. (2017) considering that being (s,a) part of the good set $L_k$, then we are assuming (Appendix E.3 Dann et al. (2017)) that $w_k(s,a) \geq w_{min}$.

(e) substituting (f) we get the upper bound, and we conclude the proof.

$\square$

**Lemma 7.2.** *Considering a non-stationary MDP M with a set of options as an SMDP $M_{\mathcal{O}}$ (Sutton et al., 1999). In $M_{\mathcal{O}}$ the number of decisions taken in the $k^{th}$-episode is a random variable d and*

$$\sum_{i \in H} \sum_{(s,o) \in L_k} w_k(s_i, o_i, h_i) \mathbb{1}\{h_i < H\} = d \text{ with } \{\forall k : d \leq H\}$$

*Therefore, the following holds true:*

$$\sum_{k} \sum_{i \in H} \sum_{(s,o) \in L_k} w_k(s_i, o_i, h_i) \sqrt{\frac{1}{n_k(s_i, o_i, h_i)}} = \tilde{O}\left(\sqrt{SOKd^2}\right)$$

*Proof.* Due to the stochasticity of the option's duration, $d$ is a random variable expressing the number of decisions taken in a step. Thus, first, we can rewrite passage (*b*) of the proof of lemma 17 Zanette & Brunskill (2018) then, we change lemma E.1 considering the same notion of good set considered in the appendix of Zanette & Brunskill (2018) and the validity of lemma 2 of Zanette & Brunskill (2018), in the options framework(replacing *o* with *a*). If all the aforementioned assumptions hold, thus the derivation of the new lemma follows the derivation of lemma E.1 $\square$

**Lemma E.2** (lemma 16 (Zanette & Brunskill, 2018) for MDPs with options of fixed lenght)**.** *For an MDP with $O$ options, with a fixed lenght $\bar{\tau}$, where the horizon is divided in $N = \frac{H}{\bar{\tau}}$ decision steps, the following holds true:*

$$\sum_k \sum_{n \in N} \sum_{(s,o) \in L_k} w_{nk}(s,o) \sqrt{\frac{1}{n_k(s,o)}} = \tilde{O}\left(N\sqrt{SOK}\right)$$

*Proof.* In this MDP the control returns to the hierarchical policy after exactly $\bar{\tau}$ time steps (the length of an option), thus, we can have at most $N = \frac{H}{\bar{\tau}}$ actions in the horizon $H$. For this reason, passage (b) of the proof of lemma E.1 become

$$\sum_{n \in N} \sum_{(s,o) \in L_k} w_{nk}(s,o) = N$$

The rest results for the same passage of the proof of lemma E.1. $\qquad \square$

To have a more complete analysis we need also to consider the triples (s, o, h) which aren't inside the good set. To do that, we can adapt Lemma 3 of Zanette & Brunskill (2018), for the FH-SMDP setting.

**Lemma E.3** (Outside the good set)**.** *It holds that:*

$$\sum_{k=1}^{K} \sum_{h=1}^{d} \sum_{(s,o,h) \notin L_k} w_k(s,o,h) = \tilde{O}(SOd)$$

The proof follows from the one of lemma 3 of Zanette & Brunskill (2018).

