# OpenReview forum: "An Option-Dependent Analysis of Regret Minimization Algorithms in Finite-Horizon Semi-MDP"
_TMLR — Accepted by TMLR_

### Review · Reviewer_tVm5 · 2023-07-04

**Summary Of Contributions:**

This paper extends the UCRL algorithm to an 'option-dependent' setting where the MDP with a longer horizon can be decomposed to several short-horizon MDPs. The authors provide a $\sqrt{T}$ regret bound for different options for these short-horizon MDPs. The learning process of these options is also discussed in the latter part of this paper, suggesting an $\sqrt[3]{T^2}$ regret bound.

**Audience:**

Yes

**Broader Impact Concerns:**

I don't have any broader impact concerns regarding this paper.

**Claims And Evidence:**

Yes

**Requested Changes:**

- I would suggest the authors formally define $d$ in the problem settings since it is also heavily discussed in Sec 5.

I also have some questions besides the above suggestions.

- Regarding $d$, if the expectation is an empirical expectation (i.e. $dT$ = Number of Decisions over $T$ episode), will it lead to some dependency issue in analyzing the regret bound?

- Why Extended value iteration is necessary, can we replace it with dynamic programming starting from $h = H$ to $h = 1$ as UCRL paper?
- What if Assumption 6.1 cannot guarantee $\pi^*(s) = \pi_o^*(s)$, which means some misspecification exists on the designed rewards and desired behavior?

**Strengths And Weaknesses:**

Strengths:

- This paper is well-written and easy to follow. The scope of this paper covers the HRLs from using the deterministic options to learning these options.
- The regret bound (both $\sqrt{T}$ for the known option setting and $\sqrt[3]{T^2}$ for the learning paradigm aligned with my intuition and seems to be correct

Weakness:

- The definition of $d$, the expected number of decisions taken in one episode, is confusing. What is the policy the expectation is taken on?

- I'm confused about how Algorithm 3 will corporate with Algorithm 1. For example, since the length of each option is defined by the probability $\beta_o$, how will the horizon length $H_o$ be calculated? Second, it seems to me that the agent needs to know the $S_o$ and $A_o$ for each option, which might be a little bit restrictive.

---

> ### Author Response · Authors · 2023-07-19
> **Rebuttal**
>
> First of all, we thank the reviewer for the comments. The modification asked are answered in what follows, and the revised version of the paper will be uploaded after all reviewers' revisions.
> - *How Algorithm 3 will corporate with Algorithm 1?*
>
>     Each options policy will be learned individually for $K_o$ episodes using Algorithm 3, which solves each option as a single FH-sub-MDP with horizon $H_o$. Then, once all the policies have been learned, they can be used as a fixed set of options in Algorithm 1 which solves the original FH-SMDP with horizon H.
>
> - *How does $H_o$ is calculated from $\beta_o$*
>
>     Considering an option $o$ with termination condition $\beta_o$ and initialization set $\mathcal{I}_o$, we can compute $h_\{o, start}$ and $h_\{o, end}$, respectively, the minimum horizon-step in which an option can start and the maximum horizon-step in which an option can end as:
>     \begin{align*}
>        h_\{o,start} = \min (h \in [H]: \forall s \in \mathcal{S},\ (s,h) \in \mathcal{I}_o), ~
>        h_\{o,end} = \max (h \in [H]: \forall s \in \mathcal{S},\ \beta_o(s,h) > 0 ), ~
>        H_o = h_\{o, end} - h_\{o,start}
>     \end{align*}
>
> - *It seems to me that the agent needs to know the $S_o$ and $A_o$ for each option, which might be a little bit restrictive*
>
>     The agents for the options learning phase must find each option policy by solving each individual sub-MDP $M_o$, defined by the corresponding action and state spaces, $\mathcal{S}_o$ and $\mathcal{A}_o$. However, the algorithm doesn't require knowing $S_o$ and $A_o$ the cardinality of the spaces; potentially, in the worst case, these spaces could be as big as the spaces defining the original MDP.
>
> - *I would suggest the authors formally define d in the problem settings since it is also heavily discussed in Sec 5.*
>
>     Thank you for the suggestion; we will introduce the definition of $d$, in Section 3 where we define the FH-SMDPs. In the revision, we will define $d$ as the average per-episode number of options that are selected in an episode of horizon H. Formally,
>     \begin{align}
>         d = \frac{1}{K} \sum_{k=1}^K \sum_{o \in \mathcal{O}} \sum_{s \in \mathcal{S}} \sum_{h \in H} n_k(s,o,h)
>     \end{align}
>     where $n_k(s,o,h)$ is the number of times an option $o$ has been selected in state $s$ in the step $h$. It is important to notice that $d$ is a random variable because even if the policy selecting among options is defined, the other uncertainty source is the option duration, which depends on the $\beta$ of the relative option.
>
> - *Regarding d, if the expectation is an empirical expectation (i.e. dT = Number of Decisions over T episodes), will it lead to some dependency issue in analyzing the regret bound?*
>
>     No, it doesn't; we are considering a high-probability analysis on the regret in which we leave this term in the upper bound formulation. A very similar analysis has been conducted in Fruit 2017, where the term $n$ is the number of decisions taken up to time T, and it is equivalent to $dK$.
>
> - *Why Extended value iteration is necessary, can we replace it with dynamic programming starting from h=H to h=1 as UCRL paper?*
>
>     Extended Value Iteration is necessary for the uncertainty over the model of the environment. Thus we need to solve the extended problem in which the reward function and transition kernel are the best possible ones contained in the confidence sets. However, we need to adapt the original version of EVI (jaksch 2010) for the FH setting and SemiMDP, where we have multi-step transitions.
>
> - *What if Assumption 6.1 cannot guarantee $\pi^*(s) = \pi^*_o(s)$, which means some misspecification exists on the designed rewards and desired behavior?*
>
>     If the assumption is violated, we have no guarantees that by optimizing the problem in the two separate phases, we will optimize the same objective for the joint optimization of both high-level and low-level policy. The learning phase could converge to a different joint optimal policy.

---

### Review · Reviewer_GasD · 2023-07-13

**Summary Of Contributions:**

This work is motivated by hierarchical reinforcement learning (RL), where a learner seeks to combine existing knowledge on how to solve a variety of low-level tasks into a high-level controller that solves the high-level goal. They formulate this as a Semi-MDP, where instead of having access to actions, the learner can play a given “option”, which is a low-level policy that acts for some number of steps. They propose an optimism-based algorithm in this setting, and show that it achieves $O(\sqrt{K})$ regret, with polynomial dependence on the size of the state space, option set, and horizon, as well as on how long each option runs. In addition, they consider the question of actually learning good “options”, and show an algorithm that achieves $O(K^{2/3})$ regret for learning options.

**Audience:**

Yes

**Broader Impact Concerns:**

None.

**Claims And Evidence:**

Yes

**Requested Changes:**

- Some discussion should be given on how the DEC and bounds from [1] would scale here. I believe the setting proposed here would fit into the interactive decision-making framework of [1], so that may immediately give a characterization of the problem.
- Some discussion on the optimal dependence on $d$ and $\tau$ would be helpful (i.e. showing a lower bound on these parameters).
- Is Assumption 2.1 actually needed given that we are in the finite-horizon setting?
- I believe it is important to construct an explicit, non-trivial example where there is a provable gain to using hierarchical RL over standard RL. That is, construct an instance or family of instances, show a lower bound on the performance of any standard RL algorithm for these instances, and apply the upper bound given in Theorem 4.1 to show that the regret of hierarchical RL is lower than the lower bound for standard RL.
- Similarly, it would greatly strengthen the paper if the discussion in section 6.1 on settings when the regret of Theorem 6.2 is lower than the regret for standard (non-hierarchical) RL algorithms could be made formal, as suggested above. As noted by the authors, comparing upper bounds does not offer much insight into which setting is actually harder, and a lower bound should instead be compared with the upper bound of Theorem 6.2.
- I was somewhat confused by the discussion on the average reward setting in Section 4.2.2. As I understand it, the algorithm given here does not handle the average reward setting. What does the sentence “On the other hand, when considering the same assumptions…” mean then? As far as I can tell the regret bound stated in that sentence is not shown to hold. I would suggest removing this section or clarifying this.
- In Theorem 6.2, how is regret defined? Furthermore, what if there are states that are not included in any $\mathcal{S}_o$?

There were several aspects of the problem setting not clearly defined:
- Is it possible for multiple options to be running at once? Or as soon as one option starts, is the algorithm unable to play additional options until it finishes?
- I could not find the definition of $V^*$, the optimal value function. I assume it is defined as the optimal value function achievable by playing only options. One could imagine a scenario where the options are simply “bad”, and one would instead prefer to simply play the raw actions. What guarantees does this algorithm have in such settings?
- A precise definition of $\tau$ should be given in the option setting.
- $d$ is only defined after Theorem 4.1, and even then is only defined informally. A formal definition of $d$ should be given.
- On the third line of section 4.2.1, the term “non-stationary” is used, but it’s not entirely clear what this means. Is this simply referring to MDPs where the transition kernel is different for each step $h$, or to settings where the transitions are changing from one episode to the next? This should be clarified.

Minor typos or wording issues which should be fixed:
- in the abstract, the first sentence should read “tasks are characterized” not “tasks is characterized”.
- there is an extra “)” in algorithm 1 on the line “Compute $n_h(s,o,h))$;”.
- I had difficulty parsing the sentence starting with “We can enhance characteristics…” at the top of page 8. Consider rewording.


[1] Foster, Dylan J., et al. "The statistical complexity of interactive decision making." arXiv preprint arXiv:2112.13487 (2021).

**Strengths And Weaknesses:**

Strengths:
- Hierarchical RL is practically relevant, but has lacked a rigorous theoretical treatment for the most part. Thus, this work fills a relevant hole in the literature.
- This should be made more explicit (see below), but the analysis in this work could likely be used to show that there are cases where hierarchical RL is formally easier than standard RL. While perhaps not too surprising, to my knowledge this has not been shown before.


Weaknesses:
- The regret bound obtained is not optimal, scaling as $O(H^{3/2} S \sqrt{A T})$ in the tabular setting (with no options).
- Assumption 6.1 seems very strong. It simply would not be the case in general that one could optimize a small sub-MDP, and that the optimal policy on that sub-MDP is equal to the optimal policy on the full MDP (for the states in the sub-MDP). Furthermore, the $O(K^{2/3})$ regret given in Theorem 6.2 (where the learner is also trying to learn options) is likely very suboptimal in this setting.

---

> ### Author Response · Authors · 2023-07-21
> **Response to reviewer GasD - part 1**
>
> First of all, we thank the reviewer for the interesting review and for all the suggested points. We will answer most of them in the following, and then we will provide a revised version of the paper in the following days.
>
> - *The regret bound obtained is not optimal, scaling as $O(H^{3/2}S\sqrt{AT})$ in the tabular setting (with no option)*
>
>     We are conscious of the suboptimality in the $S$ and $H$; however, as stated in the remark paragraph of Section 4.2.1, the goal of this work was to provide a first insight to theoretically motivate the empirical benefit of some HRL approach over standard RL ones in particular scenarios. We still think the same analysis can be done in the FH-SMDP setting to achieve a tighter bound with the same dependencies on the state cardinality and Horizon; Nevertheless, we leave these further improvements for future works.
>
> - *Assumption 6.1 seems very strong. It simply would not be the case in general that one could optimize a small sub-MDP, and that the optimal policy on that sub-MDP is equal to the optimal policy on the full MDP (for the states in the sub-MDP). Furthermore, the $O(K^{2/3})$ regret given in Theorem 6.2 (where the learner is also trying to learn options) is likely very suboptimal in this setting.*
>
>     Regarding assumption 6.1, as answered to reviewer S8Us, we agree that it seems demanding; however, our goal was not to prove that, in general, using a hierarchical approach would be better than using standard algorithms; we would like to show that in particular problems, there are theoretically proved improvements. In this manner, the assumption proposed is the first one, to the best of our knowledge, that attempts to characterize a structural property of the FH-MDPs that are suitable for being addressed by means of a hierarchical approach. Indeed, if Assumption 6.1 is violated, it means that the inner-option learning deviates from the process of learning the optimal policy in the flat MDP, possibly preventing the convergence to the optimal policy in the hierarchical architecture. An example of a problem in which such an assumption is valid is the problem presented in section 6 in the paragraph below the assumption.
>     Concerning the suboptimality of $O(K^{2/3})$, it is due to the choice of approach, Explore-then-Commit, in this case, learn the options and then use them. We agree that more efficient approaches can be used, which are left for future work. Despite this, our work is the first regret analysis that relaxes the assumption of having a set of pre-trained options, which in our opinion, is even stronger than the assumptions we made to achieve this result.
>
> - *Some discussion should be given on how the DEC and bounds from [1] would scale here. I believe the setting proposed here would fit into the interactive decision-making framework of [1], so that may immediately give a characterization of the problem.*
>
>     Thank you for the suggestion. However, it is unclear how the framework presented in [1] could relate to the setting proposed in the paper. The interactive decision-making framework is a generalization of a Bandit problem or an Online RL problem. Nevertheless, there is no reference to the concept of temporal abstraction, which is crucial when approaching HRL and, as demonstrated in the paper, is the feature attributed to the efficiency improvement in some specific MDPs. Moreover, we don't see any way of formalizing, with the proposed framework, the structure of policies that exchange control between themselves. Therefore, we don't see any need for a discussion about adapting our setting to the decision-making framework, although we are open to further clarification.
>
> - *Some discussion on the optimal dependence on $\tau$ and $d$ would be helpful (i.e. showing a lower bound on these parameters).*
>
>     Potentially $\tau > 1$ and $d > 1$, when considering respectively a case in which there is an option with unitary length and an option with a duration equal to $H$; however, we didn't think giving these lower bounds would have been much informative.

---

> > ### Author Response · Authors · 2023-07-21
> > **Response to reviewer GasD - part 2**
> >
> > - *I believe it is important to construct an explicit, non-trivial example where there is a provable gain to using hierarchical RL over standard RL. That is, construct an instance or family of instances, show a lower bound on the performance of any standard RL algorithm for these instances, and apply the upper bound given in Theorem 4.1 to show that the regret of hierarchical RL is lower than the lower bound for standard RL.\
> >     Similarly, it would greatly strengthen the paper if the discussion in section 6.1 on settings when the regret of Theorem 6.2 is lower than the regret for standard (non-hierarchical) RL algorithms could be made formal, as suggested above. As noted by the authors, comparing upper bounds does not offer much insight into which setting is actually harder, and a lower bound should instead be compared with the upper bound of Theorem 6.2.*
> >
> >     The lower bound for FH-MDPs has been derived in Osband et al. 2016 and is of the order $\Omega(\sqrt{HSAT})$. Nevertheless, a directed comparison with the upper bound on the regret suffered by our algorithm (regret given in Theorem 4.1) presents some issues. As described in section 4.2.1, remark paragraph, our algorithm is an adaptation of UCRL2, which suffers a regret that scales suboptimally in $H$ and $S$, not matching the lower bound, inefficiencies that will emerge in the comparison. For this reason, we thought that comparing the upper bounds would be much fairer, being the two algorithms analyzed with similar techniques, providing a clearer insight regarding the benefits of HRL compared to RL, which was the actual goal of the work. The same also applies to the result given in Theorem 6.2. In future works, we plan to provide a more efficient algorithm inspired by the analysis of UCBVI for which the upper bound matches the lower bound; in that case, we will consider your comment for the comparison.
> > - *Is Assumption 2.1 actually needed given that we are in the finite-horizon setting?*
> >
> >     It is correct that the first part of the assumption *"All the options terminate with probability one in finite time"*, is already fulfilled, being the setting Finite Horizon. We will restate the assumption eliminating the first part.
> >
> > - *I was somewhat confused by the discussion on the average reward setting in Section 4.2.2. As I understand, the algorithm here does not handle the average reward setting. What does the sentence “On the other hand, when considering the same assumptions…” mean then? As far as I can tell, the regret bound stated in that sentence is not shown to hold. I would suggest removing this section or clarifying this.*
> >
> >     We will clarify this point in the revised version of the paper. With "same assumptions" we meant $R_max = 1$ and bounded holding time. We present this section to provide an analogy between this Finite Horizon result and the average reward one shown in Fruit 2017; we will make it more straightforward, as suggested.
> >
> > - *In Theorem 6.2, how is regret defined? Furthermore, what if there are states that are not included in any $S_o$?*
> >
> >     The regret is expressed in its entire form in equation (12) section 7.2. On the other hand, when the same states aren't included in any $S_o$, it would be the case that it won't be possible to reach the same optimal policy when solving the problem hierarchically for the constraints imposed by the hierarchical structure. This bias term is part of the regret, even if not considered in the two theorems, but it is commented on in section 6.1, paragraph "Final Sub-Optimality remark".
> >
> > - *Is it possible for multiple options to be running at once? Or as soon as one option starts, is the algorithm unable to play additional options until it finishes?*
> >
> >     There are no constraints regarding the parallelism of option learning. It is possible to run O instances of Algorithm 3, one for each option.
> >
> > - *Definition of $V^*$*
> >
> >     We will add the definition of $V^*$ in the revised version, but it is correct to assume that the optimal value function is the value function of the optimal policy, selecting only among options.
> >
> >  - *One could imagine a scenario where the options are simply “bad”, and one would instead prefer to simply play the raw actions. What guarantees does this algorithm have in such settings?*
> >
> >     In our scenario, the high-level action space contains only the options, not the primitive actions, which could be selected only by the options policies.
> >
> >  - *On the third line of section 4.2.1, the term “non-stationary” is used, but it’s not entirely clear what this means. Is this simply referring to MDPs where the transition kernel is different for each step $h$, or to settings where the transitions are changing from one episode to the next? This should be clarified*
> >
> >     We use the term "non-stationarity" as in the literature to refer to MDPs in which the transition kernel is different for each step $h$.

---

> > ### Comment · Reviewer_GasD · 2023-08-05
> > **Response**
> >
> > Thanks to the authors for their detailed reply.
> >
> > One follow-up comment on the DEC: I believe the setting considered in this work would fit into the interactive decision-making setting of [1]. In particular, $\Pi$ would be the set of all policies which play options (rather than primitive actions), the observation space would be the set of all possible trajectories, and the model class $\mathcal{M}$ would be the set of all semi-MDPs (as defined in this work). Given any semi-MDP, playing a policy in $\Pi$ induces a valid distribution over trajectories, so this fits within the framework presented in [1]. As such, I think comparison with [1] is necessary.
> >
> > As a second follow-up, regarding the comparison between upper bounds, I believe the discussion in Section 6.1 could be replaced by comparing the upper bound of Theorem 6.2 to the lower bound of $\Omega(\sqrt{HSAT})$ would still give a result (one could still derive an equation similar to (10) for when Theorem 6.2 outperforms the lower bound, it may just be a somewhat more stringent condition). I believe this would be a valuable addition, as it makes a much more convincing case that hierarchical RL may yield improvements.
> >
> > [1] Foster, Dylan J., et al. "The statistical complexity of interactive decision making." arXiv preprint arXiv:2112.13487 (2021).

---

> > > ### Author Response · Authors · 2023-08-11
> > > **Response**
> > >
> > > Thanks for the further details provided. Regarding the request for additional comparison, we still don't consider it necessary to highlight the benefits of our work. On the other hand, we accept your comment on the comparison between the two upper bounds, and we change it in favor of a comparison against the lower bound of FH-MDPs. We are not sure if we can upload the revised version of the paper to be compliant with TMLR rules because we had just the first 2 weeks for the reviewer's responses. We will attach here the modification of section 6.1.
> > >
> > > *Given this result, it is now essential to understand if there are any situations in which such an approach could produce more benefit compared to a standard one. This would lead to defining classes of problems in which learning using a hierarchical approach almost from scratch should be preferred to a standard one.*
> > >
> > > *In order to do so, we compare the regret paid by our method with the theoretical lower bound for flat FH-MDPs with non-stationary transition probabilities, which is $\Omega(H\sqrt{SAT})$ (Auer et al., 2008; Ghavamzadeh et al., 2020). If the comparison results in favor of the hierarchical approach, it would be theoretically possible to define the characteristics of the problem that would discriminate the preferred framework.*
> > > *Let $\mathcal{R}$ be the ratio between the regret upper bound of our approach and the regret theoretical lower bound.
> > > \begin{align}
> > >     \mathcal{R} =\frac{Regret_{SMDP}}{\Omega(H\sqrt{SAT})} \leq \frac{K^{2/3}(H^5_O S^2_O A_O O)^{1/3}}{H^{3/2}\sqrt{SAK}}
> > > \end{align}
> > > By considering particular relations between the option-MDP and the original one, where $A_O = \alpha A$, $S_O = \alpha S$ and $H_O = \alpha H$, we can rewrite this ratio as:
> > > \begin{align}
> > >     \mathcal{R} \leq \frac{K^{1/6}\alpha^{8/3}H^{1/6}S^{1/6}O^{1/3}}{A^{1/6}}
> > > \end{align}
> > > $\mathcal{R} \leq 1$ is the condition that determines that our approach is preferable over the standard one. Thus, by imposing it, we can find the maximum number of episodes for which this constraint is satisfied.*
> > >
> > > *\begin{align}
> > >     K \leq \frac{A}{\alpha^{16} HSO^2}
> > > \end{align}
> > > The problems with characteristics compliant with this equation are the ones in which a hierarchical approach should be preferred to a standard one, even when no fixed sub-tasks policies (fixed \emph{options}) are provided.*

---

### Review · Reviewer_S8Us · 2023-07-16

**Summary Of Contributions:**

This paper studies the problem of reinforcement learning in finite horizon MDPs with options. The purpose of options is generally to handle longer sequential tasks that can be robustly divided into manageable sub-tasks handled by sub-policies. The sub-policies are managed by an overarching options policy. This paper is primarily theoretical, building on some initial work in this direction and developing theory for the finite horizon setting.
The first major contribution specializes to the setting where a set of options are fixed and given. An algorithm and regret bound are given. The regret depends only on the option-specific parameters such as the number of options and number of option decisions made in rollouts
The second major contribution focuses on learning the options simultaneously. This is given as a two-phased approach.


**Audience:**

Yes

**Claims And Evidence:**

Yes

**Requested Changes:**

The requested changes will depend on the answers to the above questions, but here are also some concrete ones.

- The regret in equation 1 is defined with value functions, which have not been defined yet. I think these are important to define first. For example, it is not clear if V* is the value function of the true optimal policy or the optimal meta-policy that optimally uses the available options. These could be different.

- More formal and detailed explanation of the two-phased algorithm and what is being assumed about access to the MDP. It seems it is implicitly being assumed that one can solve the option subproblems directly.

- More direct comparison with UCRL in discussion of the second result, as described above.

**Strengths And Weaknesses:**

Strengths:

- The paper studies an important and practically relevant problem that has not so much theoretical development so far despite its practical import.

- The technical results largely appear to be sound.

- There is helpful discussion and interpretation of the results.

- The separation between new contributions and use of existing tools is well-written, making the paper easy to understand on this front.

Weaknesses:

Minor

- The first result is not easy to interpret in my opinion, but this is eventually cleared up with the discussion on the parameter $d$ and its bound. However, in (and before) the theorem, $d$ is not formally defined. There is just a natural language description as the number of decisions in an episode. But the number of decisions of which policy? The bound is on the regret, which is defined over $K$ policies, so presumably this value is different for each policy. Is it the average over all the policies? Or is it just the one for the optimal policy?

- It is my understanding (please correct me if I am wrong) that the techniques used to develop the results in this paper are not particularly new. However, this is not an important criterion for TMLR.

Major

- The assumption about the division of the state and action spaces into regions where the options coincide with the optimal policy and the assumption that they can be individually learned is definitely strong. It is hard to imagine such an example in practice. While the first could still be an effective model, the second is key for the two-phase algorithm. Overall, in this section there is a lack of clarity about what formally the two-phased algorithm is – only the subroutine about learning the options is provided. It seems to suggest that there is an implicit assumption that one can sample an initial state for each option to learn the optimal policy, but this doesn’t appear to be explicitly stated anywhere either.


- I do not believe the second result is being demonstrated in the most understandable way. First, the UCRL $\sqrt{K}$ bounds can be easily converted to the $K^{2/ 3}$ style bounds by online-to-batch conversion. This will give a more directly comparable result for the ratio calculation: $K^{2 / 3} (H^4 S^2 A)^{1  /  3}$.

That being said, I am skeptical about whether regret is even the best way to compare them. Considering the regret bound is already compromised with $K^{ 2 / 3}$, it may be better to just consider this entirely a PAC result. In this case, to get $\epsilon$ suboptimality, UCRL will require $H^4S^2 A /\epsilon^2$ samples and the options algorithm will require $\alpha^8 H^5 S^2 A O  /\epsilon^2 +  H^2 S^2 d^2 O / \epsilon^2$ This seems to be an easier comparison. By the way, in Eq 8, should this be $\alpha^{8/3}$ instead?

---

> ### Author Response · Authors · 2023-07-20
> **Response to reviewer S8Us**
>
> First of all, we thank the reviewer for the effort and for the interesting questions. We will address all the main issues in the following, and then we will upload an updated version of the paper, with all the requested changes, in the next few days.
>
> **Minor comments:**
> - *First point*
>
>     We apologize for the confusion; we will state the definition of $d$ in the revised version. $d$ is the average per-episode number of options that are selected in an episode of Horizon H. Formally,
>     \begin{align}
>         d = \frac{1}{K} \sum_{k=1}^K \sum_{o \in \mathcal{O}} \sum_{s \in \mathcal{S}} \sum_{h \in H} n_k(s,o,h)
>     \end{align}
>     where $n_k(s,o,h)$ is the number of time an option $o$ has been selected in state $s$ in the step $h$.
>
> - *Second point*
>
>     The theoretical analysis extends the standard techniques used in Finite Horizon setting to consider temporally extended actions. The main critical point is related to the fact that d is a random variable being the duration of an option, a random variable itself. Adapting the bound on the renewal function to this setting is a novel contribution.
>
> **Major comments:**
>
> - *The assumption about the division of the state and action spaces into regions where the options coincide with the optimal policy and the assumption that they can be individually learned is definitely strong. It is hard to imagine such an example in practice. While the first could still be an effective model, the second is key for the two-phase algorithm.*
>
>     We agree that the assumption seems demanding; however, to the best of our knowledge, it is the first one that attempts to characterize a structural property of the FH-MDPs suitable for being addressed by a hierarchical approach. Indeed, if Assumption 6.1 is violated, it means that the inner-option learning deviates from the process of learning the optimal policy in the flat MDP, possibly preventing the convergence to the optimal policy in the hierarchical architecture. We provided an example of a problem in which such an assumption would be valid after the assumption itself.
>
> - *Overall, in this section, there is a lack of clarity about what formally the two-phased algorithm is – only the subroutine about learning the options is provided. It seems to suggest that there is an implicit assumption that one can sample an initial state for each option to learn the optimal policy, but this doesn’t appear to be explicitly stated anywhere either.*
>
>     Algorithm 3 is correctly a sub-routine to learn for $K_o$ episodes the optimal policy of every $o$ option. An initial state can indeed be sampled from the option's initial state distribution to start the learning procedure. Once the procedure is run for every option, it is possible to use Algorithm 1 as is, with the set of fixed options being the options with policies the optimal policy learned in the option's learning phase. We will clarify this point in section 6.
>
> - *I do not believe the second result is being demonstrated in the most understandable way. First, the UCRL $\sqrt{K}$ bounds can be easily converted to the $K^{2/3}$ style bounds by online-to-batch conversion. This will give a more directly comparable result for the ratio calculation: $K^{2/3}(H^4S^2A)^{1/3}$*
>
>     If we understand correctly, you suggest comparing our result against an adaptation of UCRL2 in an Explore-Then-Commit-like implementation. However, while it is possible to highlight much more benefits in this way, we think it wouldn't be a fair comparison, being UCRL2 not analyzed in its efficient form. Moreover, even if, as you highlighted, we are suboptimal of an order of $\sqrt{K}$, with respect to UCRL2, we demonstrated that for some MDP, we can still have benefits, even when no options are provided, that was our second goal within this paper.
>
> - *I am skeptical about whether regret is even the best way to compare them*
>
>     The regret measures it's a common measure of the performance of an algorithm in the literature of provably efficient RL methods. Moreover, the other few works which try to theoretically motivate the benefit of HRL use the same approach for the comparison (Fruit and Lazaric 2017, Fruit 2017). Therefore, we decide to follow the same procedure.

---

### Author Response · Authors · 2023-07-25
**Revised Version Uploaded**

We just wanted to notify you that we just uploaded the revised version of the manuscript with all the requested changes. Feel free to open further discussions if your questions were not answered after the posted comments.

---

### Decision · Action_Editors · 2023-08-30

**Recommendation:** Accept with minor revision

**Comment:**

This paper provides a theoretical analysis of hierarchical reinforcement learning, more specifically of options, under the setting of finite-horizon semi-MDPs. The authors extend UCLR to this setting, considering both the cases where the option are given or learnt beforehand, and provide a regret analysis, discussing the cases where options can be preferable to a flat policy.

The rebuttal and discussion mostly addressed reviewers initial concerns and questions, who are all on the acceptance side, noticing that this contribution fills a relevant hole in the literature. I thus recommend acceptance.

There was some further discussion among reviewers about the connection to Foster et al paper (cf review of GasD and associated rebuttal). It was agreed that if HRL could fit this general framework, doing so is not direct and would be a contribution by itself, so a precise comparison is out of scope. However, the authors are encouraged to mention this work, eg as a possible alternative for the theoretical study of HRL in future works. For the camera-ready, please also add the content to sec. 6.1 as given in your last reply to reviewer GasD.

**Audience:**

Yes

**Claims And Evidence:**

yes